# H3T: Efficient Integration of Memory Optimization and Parallelism for High-Throughput Transformer Training

**Yuzhong Wang**[*1], **Xu Han**[*†1], **Weilin Zhao**[*1]
**Guoyang Zeng**[2], **Zhiyuan Liu**[†1,3], **Maosong Sun**[†1,3]
[1] NLP Group, DCST, IAI, BNRIST, Tsinghua University, Beijing, China
[2] ModelBest Inc., Beijing, China
[3] Collaborative Innovation Center for Language Ability, Jiangsu Normal University, China
{yz-wang21,zwl23}@mails.tsinghua.edu.cn
zengguoyang@modelbest.cn    {hanxu2022,liuzy,sms}@tsinghua.edu.cn

## Abstract

In recent years, big models based on Transformers have achieved state-of-the-art performance on many artificial intelligence (AI) tasks. Despite the success of these Transformer-based models, their huge parameter size poses a serious challenge to their training, both from the storage and computation perspectives. To this end, memory optimization (e.g., rematerialization and offloading) and parallelism (e.g., data parallelism and model parallelism) are widely explored to make training Transformers more efficient. In this paper, we propose a framework to automatically find an efficient integration of memory optimization and parallelism for **H**igh-**T**hroughput **T**ransformer **T**raining (named **H3T**), which is rarely considered by existing efforts for training big Transformer-based models. Specifically, we design search algorithms to combine appropriate memory optimization strategies and parallelism schemes to achieve a balance between memory overhead and training efficiency. We implement H3T based on an open-source toolkit BMTrain and then use H3T to train the Transformers of different sizes to evaluate the efficiency of H3T. The experimental results show that H3T outperforms the most popular deep learning (DL) toolkit Megatron-DeepSpeed by $1.2\times \sim 4.3\times$ training speed while reducing $34.6\% \sim 80.5\%$ of memory overhead. Moreover, H3T can use only 64 NVIDIA A100 GPUs to train GPT-3-175B, which is very difficult for existing DL toolkits. The source code is available at `https://github.com/OpenBMB/BMTrain/tree/h3t`.

## 1 Introduction

In recent years, the emergence of Transformers [47] has significantly advanced the development of the AI field. Owing to the strong abilities of sequence modeling and transduction brought by the attention mechanisms, Transformer-based models have achieved state-of-the-art performance on many tasks and have become the foundation architecture for various AI directions, such as natural language processing [33, 9, 34, 4], computer vision [10, 23, 22], and multimodal processing [32, 39, 38].

Despite the outstanding performance, with the trend of continuously growing model size, the computation and storage costs become a severe challenge for training most big Transformer-based models. For example, the popular big model GPT-3 [4] has 175 billion parameters, and all these parameters require at least 700 GB of memory to store in `float32` format, not to mention the forward activations,

---

[*]Indicates equal contribution.
[†]Indicates corresponding authors.

37th Conference on Neural Information Processing Systems (NeurIPS 2023).

backward gradients, and optimizer states. Even ignoring the memory overhead, according to the existing analysis [28], it will take about 288 years to train GPT-3 with a single NVIDIA V100 GPU.

To tackle these issues, researchers explore various memory optimization strategies and parallelism techniques. Memory optimization strategies such as offloading [42, 6, 16, 2, 41, 37, 45] and rematerialization [12, 5, 20, 18, 19] aim to reduce the memory overhead of one (or each) single GPU. Parallelism techniques such as data parallelism [53, 30, 11] and parameter parallelism [8, 17, 27, 51, 36] mainly focus on improving both training efficiency and memory utilization across multiple GPUs. Some hybrid memory optimization strategies [49, 31, 3, 15] and automatic parallelism techniques [52, 46, 26] have also been proposed to further improve the storage and computation. Although promising results have been achieved on both tracks of memory optimization and parallelism, respectively, very limited efforts have been devoted to combining them both in an easy and efficient way.

In this paper, we propose a framework to automatically find an efficient integration of memory optimization and parallelism for **H**igh-**T**hroughput **T**ransformer **T**raining (named **H3T**), which is rarely considered by existing efforts for training big models. In H3T, we formalize memory optimization strategies (including offloading and rematerialization) and parallelism techniques (including data parallelism, parameter parallelism, and zero redundancy optimizer (ZeRO)) into multiple optimization switches. We then arrange an automatic solver that selects appropriate switches at each Transformer layer, while the core purpose of the solver is to achieve better training efficiency under the memory constraints. Since each transformer layer has a set of optimization switches independent of the other layers, the giant search space poses a challenge for the solver. Toward this end, we introduce greedy and dynamic programming (DP) algorithms to the searching process, enabling the solver to consider both search performance and search efficiency.

In experiments, our simulation studies show that H3T with our designed solvers outperforms traditional manual optimization without H3T. We also implement H3T based on BMTrain, an open-source toolkit that accelerates the training of big Transformers with various parallelism techniques. We conduct actual-running experiments to compare H3T and Megatron-DeepSpeed [40], one of the most popular distributed learning toolkits for DL. The results show H3T can train big Transformers $1.2\times \sim 4.3\times$ faster than Megatron-DeepSpeed while reducing the memory usage by $34.6\% \sim 80.5\%$. Moreover, supported by our implementation of H3T, it is possible to train GPT-3 with 175 billion parameters by using 64 NVIDIA A100 GPUs.

We summarize our contributions as follows: (1) We design H3T, which is, to our best knowledge, the first to automatically integrate memory optimization and parallelism for training big Transformers. (2) We conduct adequate experiments that show H3T outperforms conventional manual optimization as well as the most popular DL toolkit Megatron-DeepSpeed. (3) Supported by our implementation of H3T, we can train GPT-3 with 175 billion parameters on 64 NVIDIA A100 GPUs.

## 2 Background and Related Work

### 2.1 Transformer-based Model

Transformer [47] is a widely-used deep neural network architecture. Due to the powerful capability of modeling data, the Transformer architecture is used to build various pre-trained models. These pre-trained models can capture rich knowledge from massive unlabeled data and then transfer the captured knowledge to handle various complex AI tasks. Transformer-based pre-trained models have become one of the cornerstones of recent AI developments, especially in the NLP domain. Currently, Transformer-based models achieve state-of-the-art performance on almost all NLP tasks [9, 21, 35, 33, 34, 4]. Inspired by the success of these models in the NLP domain, Transformers are also explored in other important AI domains like CV [10, 23, 22] and Multi-Modal (MM) [32, 39, 38, 29].

Despite the remarkable achievements of Transformers, it is a big challenge to train big Transformer-based models. On the one hand, the parameter size of popular models is increasing in an explosive way and has already exceeded Moore's Law. Take OpenAI's GPT as an example: the parameter sizes of GPT-1 [33], GPT-2 [34], and GPT-3 [4] released in 2018, 2019, and 2020 are 117 million, 1.5 billion, and 175 billion, respectively. On the other hand, Transformers contain heavy dense algebraic operations, such as matrix multiplication, making the time and memory complexities of these operations much higher than the parameter size. The combination of rapid parameter growth and huge computational overhead challenges the DL community.

## 2.2 Memory Optimization

**Rematerialization**. Rematerialization, also known as checkpointing, is the process of discarding intermediate results within the GPU and recomputing these discarded results when needed. Rematerialization is first proposed by Grimm et al. [13] and first implemented by Griewank et al. [12]. Chen et al. [5] reduce the memory complexity of the conventional rematerialization method to $O(\sqrt{N})$ without changing the computational time complexity, which is a milestone that makes rematerialization known to researchers and implemented in many DL toolkits. However, Chen's work is based on artificial rules and cannot precisely balance memory overhead and training efficiency for different models and environments, limiting its performance and scalability. Many researchers devote their efforts to developing automatic rematerialization methods to tackle this issue. Dynamic programming (DP) [14, 20] and mixed-integer linear programming (MILP) [18] are both adopted to find better rematerialization schemes by automatically arranging which neural network layers need to be rematerialized to reduce the memory overhead while minimizing the efficiency loss.

**Offloading**. Offloading aims to transfer appropriate tensors from GPUs to CPUs and prefetch them back when they are required for computation. Preliminary offloading methods [42, 6, 44] mainly focus on using empirical rules to select the tensors for offloading. However, since offloading introduces complicated communication overhead for data transferring, it is not easy to empirically find the global optimal solution. For this reason, researchers attempt to offload tensors more intelligently. Huang et al. [16] perform tensor offloading based on a custom-designed genetic algorithm, while Beaumont et al. [2] use a DP algorithm. Besides the above offloading methods that mainly focus on activation tensors and convolutional neural networks (CNNs), some other researchers [41, 45] also explore offloading parameters and optimizer states, which can effectively reduce GPU memory usage for other types of model architectures like Transformers.

**Hybrid Memory Optimization**. Rematerialization saves memory by bringing additional tensor computation, while offloading saves memory by introducing extra tensor movement. These two optimization strategies can coexist, so integrating them and saving more memory is a natural idea. Early works [49, 31] attempt to intelligently combine them using heuristics and achieve remarkable performance. Afterward, some scholars explore using better algorithms, such as DP [3] and MILP [50], to model the integration of rematerialization and offloading and thus to find better hybrid optimization schemes. Although these hybrid methods have achieved promising results, existing works ignore incorporating them with parallelism techniques that are crucial for training big models. In view of this, we explore finding an efficient integration of memory optimization strategies and parallelism techniques for training big Transformers.

## 2.3 Parallelism

**Data Parallelism**. Data parallelism [53, 30, 11] is a typical approach for distributed DL. Its main idea is to divide the batch of training data into several mini-batches and process them on different devices (GPUs) in parallel. To synchronize the model parameters distributed on multiple devices, the gradients of parameters must be averaged among all devices before updating parameters in the parameter optimizer. Based on data parallelism, zero redundancy optimizer (ZeRO) [36] is proposed to partition the optimizer states and parameter gradients, which can significantly reduce the memory overhead of data parallelism without changing the computational complexity.

**Model Parallelism**. In recent years, the parameter size of DL models is continuously growing, and the conventional training scheme that allocates the entire model on all devices can easily run out of memory. To tackle this issue, researchers explore various model parallelism methods that distribute model parameters over multiple devices. There are generally two branches of model parallelism. One is pipeline parallelism [8, 17, 27, 51], which partitions model parameters by layer level and lets devices be responsible for the computation of different layers. The other is parameter parallelism [28, 40], which partitions parameter matrices within a layer to enable distributed computing in multiple devices. Like data parallelism, model parallelism can also be incorporated into ZeRO by partitioning the model at the tensor level and gathering the required parameters before computation. Most recent distributed DL toolkits for training big Transformers, such as BMTrain and Megatron-DeepSpeed, are designed based on data parallelism, model parallelism, and ZeRO.

**Automatic Parallelism**. Although many efforts have attempted to combine model parallelism and data parallelism to improve efficiency further, finding the optimal combination based on empirical

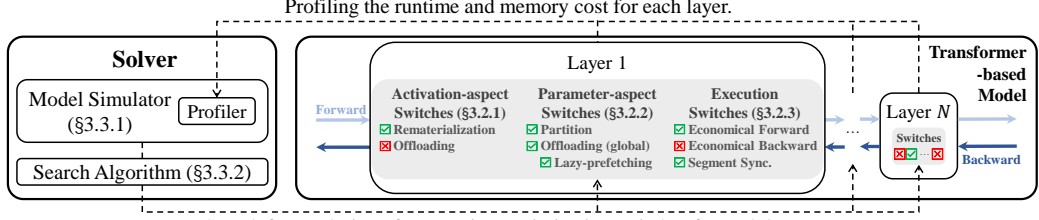

Figure 1: The overall architecture of H3T.

rules is not easy. To this end, some scholars explore automatic parallelism methods. Galvatron [26] uses a decision tree and a DP algorithm to find a hybrid parallelism solution among multiple parallelism dimensions. Alpa [52] and Unity [46] focus on finding the parallelism scheme at the computational graph level and use DP algorithms to search for the scheme based on their defined parallel operators. However, most existing works of automatic parallelism pay more attention to execution efficiency and rarely involve memory optimizations, which makes the memory overhead suboptimal for training big models. Therefore, we introduce memory optimization strategies to the existing parallelism schemes for better results.

## 3 H3T Framework

### 3.1 Formalization of Training Transformers

We first formalize the process of training Transformers. Generally, training DL models involves three stages: forward propagation, backward propagation, and parameter optimization. As a typical sequential model, Transformers can be formalized as an ordered sequence of neural layers, with each layer only connected to adjacent ones. So we go deep into the layer aspect and define $\texttt{Forward}_i, \texttt{Backward}_i, \texttt{Optimize}_i$ as the three stages at the $i$-th layer, respectively.

In the forward propagation, $\texttt{Forward}_i$ takes $\mathbf{H}_i$ and $\mathbf{P}_i$ to calculate $\mathbf{H}_{i+1}$, where $\mathbf{H}_i$ and $\mathbf{H}_{i+1}$ are the hidden states, and $\mathbf{P}_i$ is the parameters of the $i$-th layer. Along with generating some intermediate activations $\mathbf{A}_i$, the forward propagation at the $i$-th layer can be formalized as

$$\mathbf{H}_{i+1}, \mathbf{A}_i = \texttt{Forward}_i(\mathbf{H}_i, \mathbf{P}_i). \tag{1}$$

The final output $\mathbf{H}_{N+1}$ is used to calculate the loss function and the initial gradient $\nabla\mathbf{H}_{N+1}$. Then, in the backward propagation, $\texttt{Backward}_i$ is to calculate the gradient $\nabla\mathbf{H}_i$ and $\nabla\mathbf{P}_i$, by using $\nabla\mathbf{H}_{i+1}$, $\mathbf{H}_i, \mathbf{P}_i$, and the intermediate activations $\mathbf{A}_i$ generated by $\texttt{Forward}_i$, which can be given as

$$\nabla\mathbf{H}_i, \nabla\mathbf{P}_i = \texttt{Backward}_i(\mathbf{H}_i, \mathbf{P}_i, \mathbf{A}_i, \nabla\mathbf{H}_{i+1}). \tag{2}$$

After the backward propagation, $\texttt{Optimize}_i$ is to use some parameter optimization functions to update the parameters $\mathbf{P}_i$ according to the gradients $\nabla\mathbf{P}_i$.

In each training step, the conventional training way is first to run $\texttt{Forward}_i$ from $i = 1$ to $N$, then run $\texttt{Backward}_i$ in reverse order, and finally use the parameter optimizer to update parameters according to gradients. For H3T, the training step is similar to the conventional one, but it is allowed to adjust the execution order or add some additional operations under the premise that the gradients are calculated correctly and the parameter optimization operations are performed stably, which can bring us much flexibility and convenience to integrate memory optimization and parallelism.

### 3.2 Optimization Switches

### 3.2.1 Activation-aspect Optimization Switches

For the sequential model, activations involve hidden states $\mathbf{H}_i$ and intermediate activations $\mathbf{A}_i$. They play important roles in model training and meanwhile consume a lot of GPU memory, especially intermediate activations [42]. As mentioned in the background, scholars have developed two major approaches, rematerialization and offloading, to reduce the memory overhead caused by activations.

Considering the properties of hidden states and intermediate activations, we adopt rematerialization to optimize the memory of intermediate activations while using offloading to save the memory of

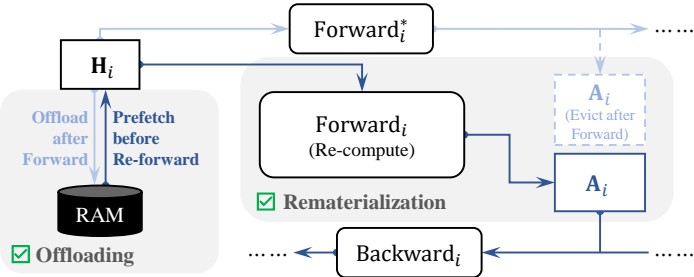

Figure 2: Activation-aspect switches.

hidden states. This setting is also applied by some existing work of hybrid optimization [3]. The two optimization switches are illustrated in Figure 2.

**Rematerializing intermediate activations** is to evict $\mathbf{A}_i$ after forward steps and recompute $\mathbf{A}_i$ before backward steps. To differentiate, we define Forward$_i^*$ as the forward computation without saving $\mathbf{A}_i$. As Figure 2 shows, rematerialization for the $i$-th layer replaces Forward$_i$ with Forward$_i^*$ in the forward stage, and insert an extra Forward$_i$ before Backward$_i$ in the backward stage.

**Offloading hidden states** is to offload $\mathbf{H}_i$ and prefetch them before the backward propagation steps. Different from intermediate activations, if we similarly try to evict and recompute hidden states, the training process will suffer from excessive recomputations, and the performance will also be affected to a great extent. To this end, offloading hidden states can be more efficient.

These two optimizations can significantly reduce the memory cost of activations. Specifically, switching on both of them for every layer will reduce the resident activation memory to $0$.

### 3.2.2 Parameter-aspect Optimization Switches

Traditional training process store all $\mathbf{P}_i$ and $\nabla\mathbf{P}_i$ on GPUs. As described above, $\mathbf{P}_i$ and $\nabla\mathbf{P}_i$ are only used for Forward$_i$, Backward$_i$, and Optimize$_i$. Therefore, it is a straightforward idea to reduce this part of memory overhead by offloading optimizer states, parameters, and gradients to CPU RAM, as shown in Figure 3a. Generally, the number of gradients equals the number of parameters, and for most common optimizers, the number of optimizer states equals or even exceeds the number of trainable parameters. Therefore, offloading them is an efficient way to reduce the GPU memory overhead. On this basis, we further implement a high-performance CPU optimizer[3] that runs Optimize$_i$ on CPUs.

Moreover, we employ parameter parallelism for H3T. Specifically, we store partitioned parameters $\mathbf{P}_i^{(j)}$ on each GPU. Before Forward$_i$, we use an all gather operation to gather them from all GPUs to get $\mathbf{P}_i$ for computation. After Backward$_i$, we evict the gathered $\mathbf{P}_i$ and use a reduce scatter operation to average and scatter the $\nabla\mathbf{P}_i$ to each GPU.

In this way, the parameters and gradients form a closed loop, as shown in Figure 3a. Due to the holistic nature and outstanding memory-saving performance, we globally turn on the above offloading optimizations and parallelism in this paper.

In addition to the above global optimizations, we have two advanced parameter-aspect switches.

**Parameter partition** (Figure 3b) is to release gathered parameters after using them and gather them before their next use. Parameter partition mainly focuses on saving the wasted memory between the forward and backward steps of each layer, which is an issue overlooked by model parallelism.

**Lazy-prefetching** (Figure 3c) is to evict the partitioned parameters after every reduce scatter operation and lazily prefetch them back before every all gather operation. Lazy-prefetching aims to eliminate the memory cost of resident partitioned parameters.

Both parameter partition and lazy-prefetching can avoid the long-term persistence of parameters during the forward-to-backward period, thus effectively saving the memory of parameters. Specifically, turning on both optional switches for every layer will reduce the resident parameter memory cost to $0$.

---

[3]The CPU optimizer is implemented based on Intel AVX instructions. More details are in the appendix.

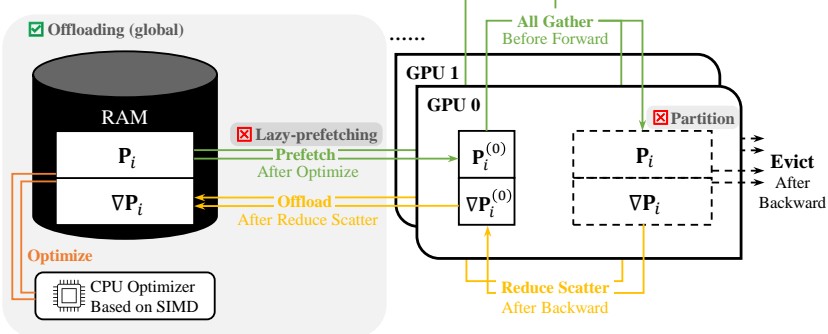

(a) Basic parameter optimization scheme based on tensor offloading and parameter parallelism.

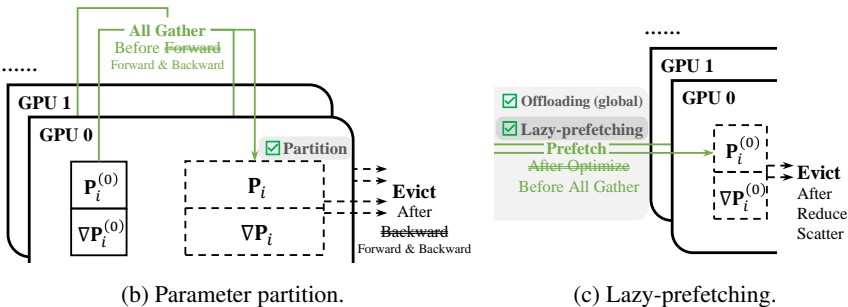

(b) Parameter partition.    (c) Lazy-prefetching.

Figure 3: Parameter-aspect switches.

### 3.2.3   Implementation: Concurrent Streams and Execution Switches

We implement the above optimizations with three concurrent streams for H3T. Specifically, we have (1) A calculation stream that takes charge of forward, backward, and parameter optimization. (2) An NVLink[4] stream that covers the GPU-GPU communication operators, i.e., all gather and reduce scatter. (3) A PCI-E stream that covers the GPU-CPU communication operators, i.e., offloading and prefetching. Following most previous works [49, 31, 3, 50], asynchronous communication and computation can overlap different parts of the time overhead, thus minimizing the efficiency loss. As illustrated in Figure 4, the main idea of our implementation is to overlap the post-process (scattering and offloading) of the previous layer and the preparation (gathering and prefetching) of the next layer with the computation of the current layer, and we call these overlapped operations a "segment".

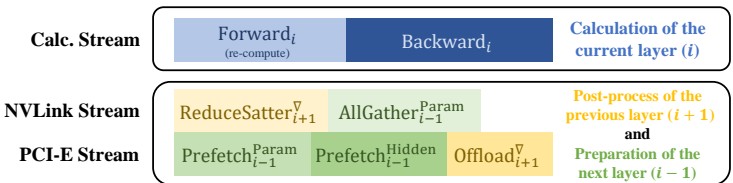

Figure 4: An illustration of backward segment $i$. Without loss of generality, we suppose the switches of layer $i-1$ and $i+1$ are all switched on.

Furthermore, we design three execution switches for H3T. **Economical forward** and **economical backward** prioritize the scattering and offloading of the previous layer in order to save more memory. In contrast, non-economical modes prioritize gathering and prefetching of the next layer, which may reduce the delay in the computation of the next layer. **Segment synchronization** is to force all streams to be synchronized between two segments. More details about the implementation and the three switches are introduced in the appendix.

---

[4]NVLink is a high-speed multi-GPU communication link developed by NVIDIA. More details are here.

### 3.3 Optimization Solver

The optimizations save memory while bringing performance losses. To tackle this, H3T involves a solver to decide the appropriate switches to achieve the trade-off between memory and speed.

#### 3.3.1 Problem Definition

We first state the problem definition for the solver. We define $\mathbf{s} = (s_1, s_2, \cdots, s_N)$ as an optimization sequence for the model, where $s_i \in S$ represents the optimization scheme for the $i$-th layer, while $S$ is the set of all combinations of optimization switches for one single layer. In this paper, $S$ involves all combinations of the 7 layer-level switches described above, so we have $|S| = 2^7 = 128$.

We define $\texttt{Runtime}(\mathbf{s}), \texttt{Memory}(\mathbf{s})$ as the runtime (per step) and the memory overhead (per device) given the model and the optimization sequence $\mathbf{s}$, respectively. Then, the goal of the solver is to find an optimization sequence $\mathbf{s}$ with lower $\texttt{Runtime}(\mathbf{s})$ under the memory constraint $M$ (per device):

$$\text{minimize } \texttt{Runtime}(\mathbf{s}), \text{ s.t. } \texttt{Memory}(\mathbf{s}) \leq M. \tag{3}$$

A key problem is how to precisely estimate $\texttt{Runtime}(\mathbf{s})$ and $\texttt{Memory}(\mathbf{s})$. Towards this goal, we introduce a profiler that collects the runtime and memory overhead of each operation of each layer at the first several training steps. Then we implement a model simulator to simulate the training process of each layer based on the model architecture and implementation, thus helping us estimate $\texttt{Runtime}(\mathbf{s}), \texttt{Memory}(\mathbf{s})$. Besides, the model simulator can also help the solver evaluate any solutions or sub-solutions when exploring different optimization schemes.

How to pick appropriate switches is another big challenge. The optimization of different layers can be in any combination, so the number of global solutions can be up to $|S|^N$. It is not easy to find a good solution from such exponential search space with the memory constraint and toward the goal of minimizing the training runtime. Toward this goal, we design three search algorithms in this paper.

#### 3.3.2 Search Algorithms

**Random solver**. Brute force does not work for our task because of the exponential search space, so we design a simple method based on random sampling to find a suboptimal solution more efficiently. Specifically, we randomly sample several optimization sequences from $S^N$ and pick the best one.

**Greedy solver**. The random solver is simple but not effective for large-scale models. The main reason is that the search space grows exponentially with $N$, but the sampling time is constant, which means the probability we hit a good solution becomes much lower with the growth of the model size. To address these issues, we design a greedy solver to search for better solutions. The core idea of our greedy solver is first assuming all optimization switches are turned on, then greedily turning off some switches to increase memory utilization and improve training efficiency. Due to the space limitation, detailed descriptions of the greedy solver are in the appendix.

**Dynamic Programming (DP) Solver**. Our task strives for better runtime under the memory constraint, similar to the knapsack problem [25, 7] that pursues higher value under the weight constraint. Inspired by the classic dynamic programming (DP) algorithm for the knapsack problem [1, 43], we design a search algorithm based on DP to solve our task better. Specifically, our DP algorithm considers the optimal sub-solutions for specified prefixes and memory limits. When transferring, we need to enumerate the next layer's switches and estimate the next segment's running status.

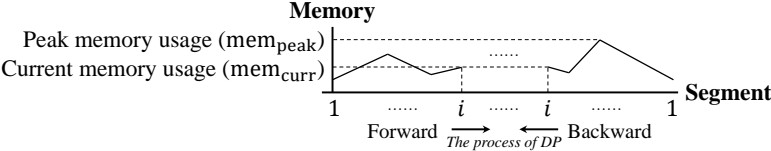

Figure 5: An illustration for $i, \text{mem}_{\text{curr}}, \text{mem}_{\text{peak}}$.

Formally, the DP state can be represented by the following variables: (1) $i$: The prefix length, which means this state is considering segment $1 \sim i$ (including forward segment $1 \sim i$ and backward segment $i \sim 1$, similarly hereinafter). (2) $\text{mem}_{\text{curr}}$: The memory usage after forward segment $1 \sim i$.

(The memory usage before backward segment $i \sim 1$ is essentially equal and can be easily calculated.) (3) $\mathtt{mem}_{\mathrm{peak}}$: The peak memory usage during segment $1 \sim i$. (We have to record $\mathtt{mem}_{\mathrm{peak}}$ because the resident memory may increase due to non-lazy-prefetching, leading to an increase in the peak memory from the previous segments.) (4) $s_i, s_{i+1}$: The switches of layer $i$ and $i + 1$, which is used to simulate segment $i + 1$.

We use $\mathtt{sol}(i, \mathtt{mem}_{\mathrm{curr}}, \mathtt{mem}_{\mathrm{peak}}, s_i, s_{i+1})$ to represent the optimal sub-solution of the state. Then we can enumerate $s_{i+2}$ to transfer to the next state $(i + 1, \mathtt{mem}'_{\mathrm{curr}}, \mathtt{mem}'_{\mathrm{peak}}, s_{i+1}, s_{i+2})$. During the transfer, we need to: (1) Simulate the forward segment $i + 1$ and backward segment $i + 1$ based on $s_i, s_{i+1}, s_{i+2}$. (2) Calculate $\mathtt{mem}'_{\mathrm{curr}}, \mathtt{mem}'_{\mathrm{peak}}$ and the total runtime based on $\mathtt{mem}_{\mathrm{curr}}, \mathtt{mem}_{\mathrm{peak}}$, the previous runtime, and the simulation result. (3) Update $\mathtt{sol}(i + 1, \mathtt{mem}'_{\mathrm{curr}}, \mathtt{mem}'_{\mathrm{peak}}, s_{i+1}, s_{i+2})$ if the peak memory $\mathtt{mem}'_{\mathrm{peak}}$ does not exceed $M$ and the runtime is shorter than the existing solution's.

There are two noteworthy details for our DP algorithm. First, considering the overlapping sub-problems condition, we must ensure the state is finite, so we discretize the memory limit into $m + 1$ values that are evenly distributed in $[0, M]$. Second, following the condition of optimal substructure, the problem for each state must be reducible to the corresponding sub-problem for the other states. In this paper, only cross-segment communications violate this condition, so we force segment synchronization switched on and remove it from the switch set (thus $|S|$ reduce to 64). These two assumptions may affect the result but make our algorithm design much simpler and more efficient. We leave the design of the more comprehensive DP algorithm as our future work.

In addition to the basic version of the DP algorithm, we have two prunings to improve its efficiency. Please refer to the appendix for them along with the pseudo-code and time complexity analysis.

## 4 Experiments

### 4.1 Experimental Settings

**Models**. We use BERT[9], one of the most common Transformer-based language models, as an example for our experiments. We customize different BERT sizes for our experiments, including 1.8B, 6B, 13B, and 100B. Traditional small models, such as BERT-base (0.1B) or BERT-large (0.3B), are not considered in this paper because they can be easily trained using conventional approaches.

**Environments**. We test the performance of H3T in three different experimental environments: (1) 1 node of $8 \times$ NVIDIA GeForce RTX 2080 Ti 10 GB ($8 \times$ 2080Ti); (2) 1 node of $8 \times$ NVIDIA A100-SXM4-40GB with NVLink 3.0 ($8 \times$ A100); and (3) 8 nodes of $8 \times$ A100 ($64 \times$ A100).

**Implementations**. We implement H3T based on BMTrain, an open-source toolkit that accelerates the training of big Transformers with various parallelism techniques.

**Baselines**. Except for H3T with random solver and greedy solver, we introduce manual optimization strategy without H3T and another toolkit Megatron-DeepSpeed for comparison. (1) In the simulation study, we test the performance of manually enumerating the combination of global optimizations and picking the best one. In other words, we simulate arbitrarily and globally turning on or off all the switches and picking the best-performing solution that does not exceed the memory limit. (2) In the actual-running experiment, we attempt to turn on all optimization switches to achieve the lowest memory overhead since testing all manual optimization schemes is time-consuming and resource-intensive. (3) We introduce one of the most popular distributed deep learning toolkits, Megatron-DeepSpeed [40], for comparison in our actual-running experiment. For a fair comparison, we test Megatron-DeepSpeed with ZeRO-2 and ZeRO-3 respectively (ZeRO-2 distributes optimizer states and gradients; ZeRO-3 distributes optimizer states, gradients, and parameters [36]). Please refer to the appendix and the code for detailed configurations of Megatron-DeepSpeed and H3T in the actual-running experiment.

We do not test POFO [3], which also uses DP to solve the memory optimization problem for big model training. We consider it unfair to compare POFO with H3T because POFO does not support any parameter-aspect optimization, such as parameter offloading and parallelism. Considering the large model size, it is almost impossible for a vanilla POFO to train the large-scale language models in this paper. If we arbitrarily implement parameter optimizations for POFO, additional communication will affect the well-designed offloading solution of POFO and seriously hurt its performance.

**Other settings**. Please refer to the appendix for other detailed experimental settings, including detailed model size, hardware configurations, training settings, etc.

## 4.2 Simulation Studies

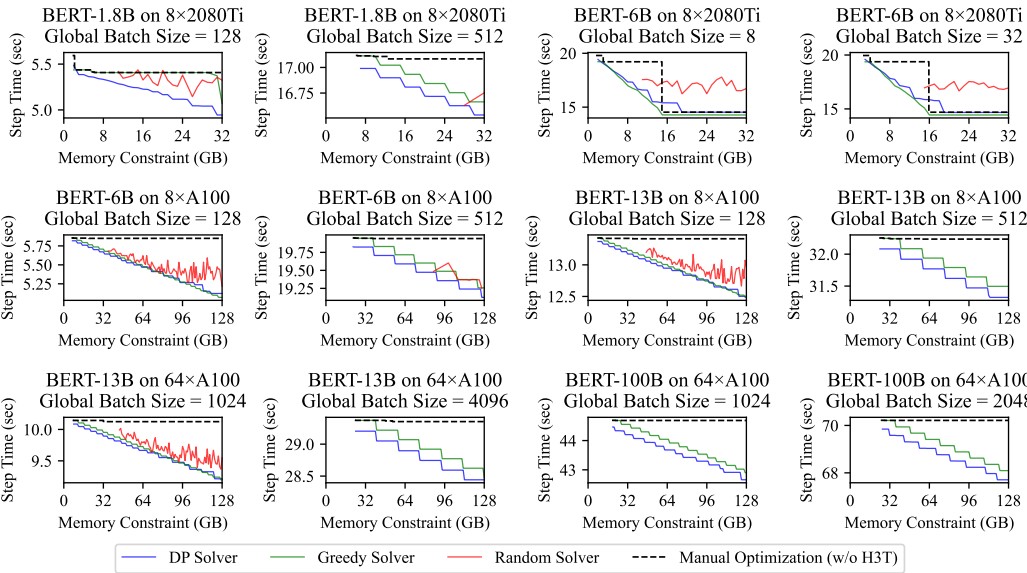

Figure 6: The results of the simulation study.

In this section, we report our simulation studies to evaluate the ideal performance of H3T. We profile the models in different environments and use the model simulator to evaluate the performance of different solvers and the manual strategy. The result is shown in Figure 6.

The result shows that our DP solver outperforms the other three baseline approaches in most cases. The greedy solver also performs well and is usually close to DP. Both DP and greedy solver have the flexibility to adjust the optimization solution based on variant memory constraints. This conclusion is applicable in all three environments, showing H3T has good adaptability to GPUs with different numbers, performance, and connections. In contrast, the random solver is less stable and less effective. Although the performance gap is not significant when memory is sufficient, the random solver, more often than not, struggles even to find an available solution.

Compared to three automatic solvers, manual optimization without H3T is often the worst, especially when memory is sufficient. This indicates that our layer-level automatic solver is able to precisely decide the optimization scheme for large-scale Transformer-based models given the memory constraint. In contrast, the traditional manual strategy performs worse and is heavy for users as well.

However, the above rules do not always stand. For BERT-1.8B with a batch size of 128 on 8×2080Ti, the greedy solver underperforms the random solver. For BERT-6B on 8×2080Ti, the DP solver continuously underperforms the greedy solver and even underperforms the manual strategy in some cases. This is because the local optimality may let the greedy algorithm fall into a suboptimum, and the segment synchronization assumption and the memory discretization may influence the DP algorithm. We argue that it is normal for such problems to occur in a few cases.

## 4.3 Actual-running Experiments

We conduct our actual-running experiment in the three environments, and the result is shown in Table 1. The results demonstrate our great advantage over Megatron-DeepSpeed. Compared to the Megatron-DeepSpeed with ZeRO-2 (ZeRO-3), H3T improves the training speed by $1.2\times \sim 1.8\times$ ($1.9\times \sim 4.3\times$) in our experiments. Comparing the results of BERT-6B and BERT-13B, both with the batch size of 128 and both on 8×A100, we find that H3T outperforms Megatron-DeepSpeed more significantly for larger model size, which means H3T is not only faster but also more adapted to bigger models.

Table 1: The time per step of the actual-running experiment (unit: second). We do not report the random solver because it fails on all these settings. '-' means the program runs out of memory.

| Environment | | 8×2080Ti | | | | 8×A100 | | | | 64×A100 | | | |
|---|---|---|---|---|---|---|---|---|---|---|---|---|---|
| Model Size | | 1.8B | | 6B | | 6B | | 13B | | 13B | | 100B | |
| Global Batch Size | | 128 | 512 | 8 | 32 | 128 | 512 | 128 | 512 | 1024 | 4096 | 1024 | 2048 |
| BMTrain | DP | 6.0 | 17.6 | **19.4** | **19.8** | **5.5** | **19.5** | 10.0 | **28.4** | **7.4** | **26.6** | **35.6** | **61.6** |
| | Greedy | **5.8** | **17.5** | 19.6 | 20.5 | **5.5** | 19.7 | **9.7** | 28.7 | 7.6 | 26.8 | 36.2 | 62.0 |
| | w/o H3T | 7.1 | 17.7 | 22.1 | 23.0 | 5.7 | 19.8 | 9.9 | 28.6 | 7.8 | 26.9 | 36.3 | 61.8 |
| Megatron -DeepSpeed | ZeRO-3 | 11.3 | - | - | - | 15.7 | - | 43.2 | - | - | - | - | - |
| | ZeRO-2 | 9.3 | - | - | - | 6.8 | - | 17.7 | - | 10.6 | - | - | - |

Focusing on the results of BMTrain, we find that H3T with DP solver and greedy solver both outperform BMTrain without H3T by up to $1.2\times$ speed up when memory is sufficient. In contrast, when memory is tight, the two automatic solvers can also enable more memory optimization and trade time performance for keeping model training. This result is consistent with the findings from the simulation study, which further verifies the flexibility of our automatic solvers. Also, similar to the simulations, the performance of DP and greedy are close, but DP still outperforms greedy a little in most cases. Besides, the random solver fails to solve for all of the settings, so we do not report its results in the table.

The memory cost of our implementation is also better than Megatron-DeepSpeed. From Table 1, we find Megatron-DeepSpeed often runs out of memory for larger models and batch sizes, which are better supported by H3T. Hence, users can train larger models with more limited hardware, e.g., train BERT-6B on $8\times2080$Ti. Not only that, **H3T can train GPT-3-175B [4] on 64×A100 with a batch size of 512 and using only about 11 GB of memory per GPU. This is really an exciting result because we would never have dreamed that a 175B-parameter model could be trained with such limited hardware, not to mention with such low GPU memory overhead.** We conduct a comprehensive memory test in the appendix and find our implementation can save $34.6\% \sim 80.5\%$ GPU memory compared with Megatron-DeepSpeed under various settings.

Besides, we verify the correctness of our implementation. (1) We fix the input data and check whether the output loss equals the release version of BMTrain during the whole training process, which justifies our incremental development based on BMTrain. (2) We do an end-to-end training experiment on 4 actual tasks of SuperGLUE [48]. The results show that H3T can achieve comparable results with PyTorch, which confirms the soundness of the whole system. The detailed experimental results on SuperGLUE are in the appendix.

We conduct three extra experiments in this paper. Due to the space limitation, we briefly introduce them here and elaborate on the details in the appendix. (1) We conduct an energy test and find that H3T is more energy-efficient than Megatron-DeepSpeed for training big Transformer-based models. (2) We test the efficiency of the solver to confirm it is not too slow. (3) We conduct a case study to show the automatically generated optimization scheme makes sense.

# 5 Conclusion

In this paper, we propose a novel framework for automatic integration of memory optimization and parallelism for **H**igh-**T**hroughput **T**ransformer **T**raining, named **H3T**. Despite the challenge of large search space, H3T significantly outperforms some manual strategies and one of the most popular DL toolkits, Megatron-DeepSpeed, in our experiments. More excitingly, H3T can use only 64 NVIDIA A100 GPUs to train Transformer-based models with more than 100 billion parameters like GPT-3-175B. We hope that the release of H3T will lower the barrier to training big Transformer-based models and promote the development of the AI community.

# Acknowledgments

This work is supported by the National Key R&D Program of China (No.2022ZD0116312), Major Project of the National Social Science Foundation of China (No. 22&ZD298), the National Key R&D Program of China (No. 2022ZD0160501), and Institute Guo Qiang at Tsinghua University. Weilin Zhao is supported by Tsinghua University Initiative Scientific Research Program.

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

# A Implementation Details

## A.1 Timing Diagram of Multiple Streams with Optimization Switches

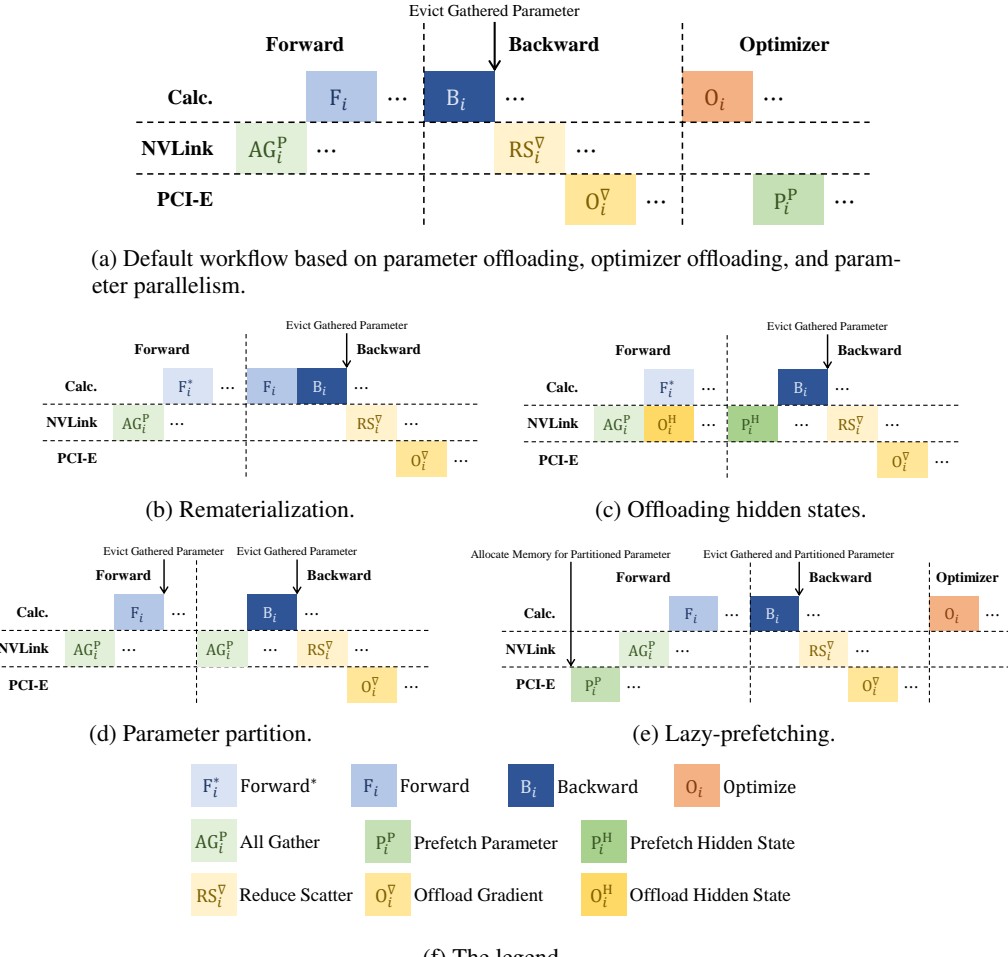

(a) Default workflow based on parameter offloading, optimizer offloading, and parameter parallelism.

(b) Rematerialization.

(c) Offloading hidden states.

(d) Parameter partition.

(e) Lazy-prefetching.

(f) The legend.

Figure 7: Timing diagram of H3T with different optimization switches.

Figure 7 illustrate the timing diagram of our implementation of H3T. Figure 7a is the default workflow based on parameter offloading, optimizer offloading, and parameter parallelism. Figure 7b and 7c are the workflow with two activation-aspect switches, respectively. Figure 7d and 7e are the workflow with two parameter-aspect switches, respectively.

## A.2 Execution-aspect Switches

In this part, we introduce the detail of our three execution switches.

**Economical forward/backward**. These two switches are to prioritize the scattering and offloading of the previous layer in order to save more memory. In contrast, non-economical forward/backward prioritize gathering and prefetching of the next layer, which may reduce the delay in the computation of the next layer. Since NVLink is not available for all GPUs, e.g., the NVIDIA GeForce series, we design the workflow with and without NVLink, respectively, and the timing diagram is shown in Figure Figure 8. It is worth mentioning that there are still asynchronous operations in the economical mode, so it is not memory-optimal but balances the efficiency.

**Segment synchronization**. Let the computation of one layer and the concurrent communications (post-process of the previous layer and preparation of the next layer) be a "segment". Then segment

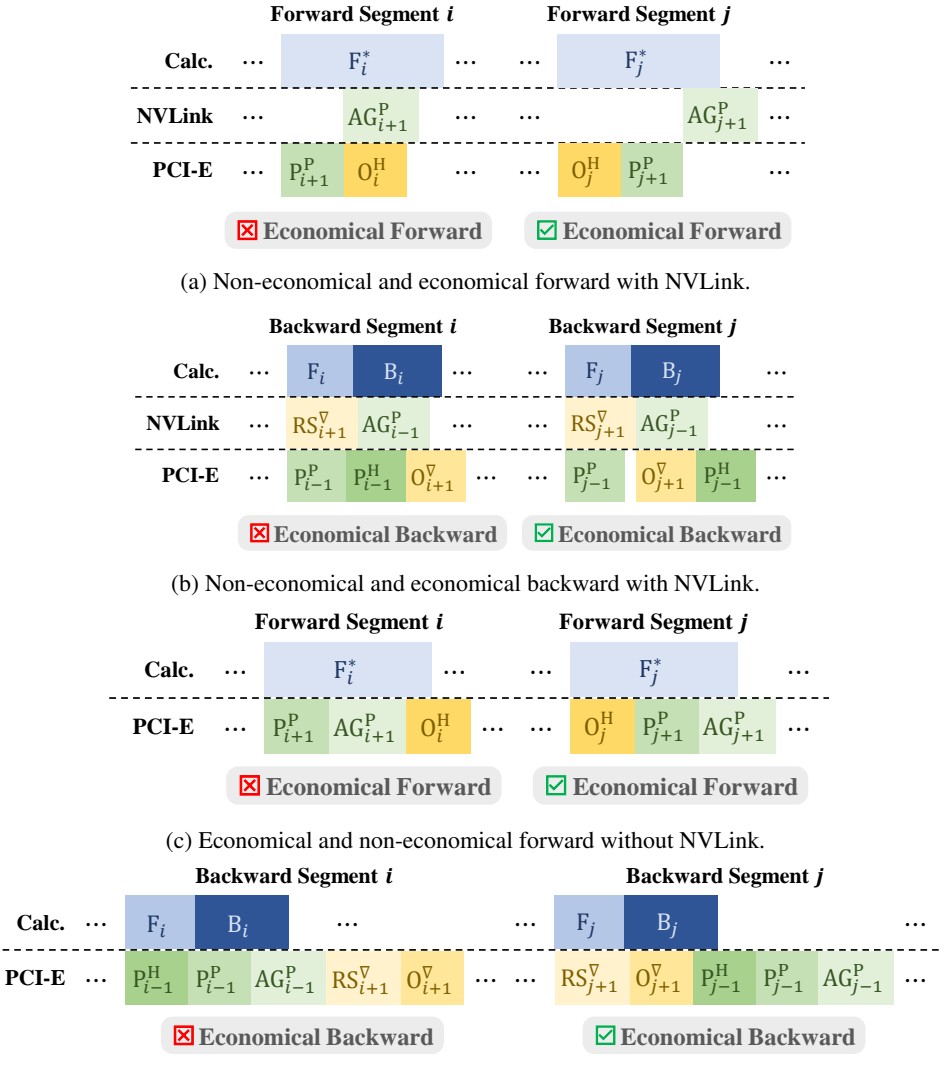

(a) Non-economical and economical forward with NVLink.

(b) Non-economical and economical backward with NVLink.

(c) Economical and non-economical forward without NVLink.

(d) Economical and non-economical backward without NVLink.

Figure 8: Timing diagrams of H3T's workflow in economical and non-economical mode. The legend is the same as in Figure 7f. Without loss of generality, we supposed all optimizations are switched on.

synchronization can be defined as: all streams need to be synchronized between two segments. In other words, if segment synchronization is on, the next segment must keep waiting until all tasks of the current segment end. In contrast, if segment synchronization is off, the next segment can start before the current stage ends. An illustration of segment synchronization is shown in Figure 9. Similar to the economical mode, segment synchronization can reduce the memory overhead, but may sacrifice a little performance. Segment synchronization is also a condition of our DP solver.

## A.3 CPU Optimizer

As we mentioned in the main text, we implement our CPU optimizer based on Intel AVX instructions. Intel AVX (Advanced Vector Extensions) instructions are SIMD (Single Instruction Multiple Data) instructions proposed by Intel for high-performance parallel computing. With the help of Intel AVX instructions, our CPU optimizer can achieve comparable performance to the conventional GPU optimizer, which greatly ensures training performance.

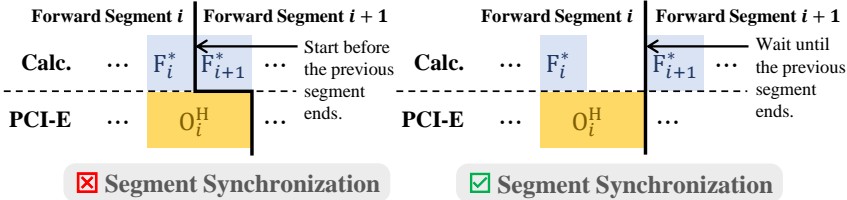

Figure 9: A schematic diagram of segment synchronization. The legend is the same as in Figure 7f.

# B  Solver Details

In this part, we mainly show the pseudo-code of the three solver algorithms and analyze their time complexity. We also describe details of the greedy solver and the pruning of the DP solver, which are not introduced in the main text due to the space limitation.

## B.1  Random Solver

---
**Algorithm 1** Random Solver

---
**Require:** $N, M, \texttt{Runtime}\,(\mathbf{s})\,, \texttt{Memory}\,(\mathbf{s})\,, T_{\text{rand}}$
 1: $\mathbf{s}_{\text{optimal}} \leftarrow \texttt{null}$
 2: **for** $i = 1$ **to** $T_{\text{rand}}$ **do**
 3:     Randomly sample an optimization sequence $\mathbf{s} \in S^N$
 4:     **if** $\texttt{Memory}\,(\mathbf{s}) \leq M$ **and** $\texttt{Runtime}\,(\mathbf{s}) < \texttt{Runtime}\,(\mathbf{s}_{\text{optimal}})$ **then**
 5:         $\mathbf{s}_{\text{optimal}} \leftarrow \mathbf{s}$
 6:     **end if**
 7: **end for**
**Ensure:** $\mathbf{s}_{\text{optimal}}$

---

The pseudo-code of our random solver is shown in Algorithm 1. Here we use a hyper-parameter $T_{\text{rand}}$ to represent the sampling times. Since the time complexity of the model simulator is $\Theta\,(N)$, the time complexity of the random solver is $\Theta\,(N T_{\text{rand}})$.

## B.2  Greedy Solver

---
**Algorithm 2** Greedy Solver

---
**Require:** $N, M, \texttt{Runtime}\,(\mathbf{s})\,, \texttt{Memory}\,(\mathbf{s})$
 1: $\mathbf{s}_{\text{optimal}} \leftarrow \texttt{null}$
 2: **for** $n_{\text{flexible}} = 1$ **to** $N$ **do**
 3:     **for** $s_{\text{flexible}} \in S$ **do**
 4:         Let $\mathbf{s}$ be the sequence that turns on all optimization switches.
 5:         **for** $i = 1$ **to** $n_{\text{flexible}}$ **do**
 6:             $\mathbf{s}\left[1 + (i-1)\left\lfloor \frac{N}{n_{\text{flexible}}} \right\rfloor\right] \leftarrow s_{\text{flexible}}$
 7:         **end for**
 8:         **if** $\texttt{Memory}\,(\mathbf{s}) \leq M$ **and** $\texttt{Runtime}\,(\mathbf{s}) < \texttt{Runtime}\,(\mathbf{s}_{\text{optimal}})$ **then**
 9:             $\mathbf{s}_{\text{optimal}} \leftarrow \mathbf{s}$
10:         **end if**
11:     **end for**
12: **end for**
**Ensure:** $\mathbf{s}_{\text{optimal}}$

---

The pseudo-code of our greedy solver is shown in Algorithm 2. We first assume all optimization switches are turned on, which reduces the memory overhead to the lowest (in our framework). Then, in order to increase memory utilization and improve training efficiency, we want to turn off some switches for part of the layers. Here we have a greedy assumption that these "partial layers" have

the same switches turned off, which can be easily enumerated as $s_{\text{off}} \in S$. Then we enumerate $n_{\text{off}}$, representing how many layers we will turn off $s_{\text{off}}$ for. Finally, we greedily put these $s_{\text{off}}$ into the initial sequence evenly.

The greedy solver introduces two greedy assumptions, where the former is centered on the idea that the best is always the best, and the latter on the idea that an even distribution may block communications less. We cannot prove that either of them is optimal, but they both make sense, so we expect our greedy solver to outperform the random solver.

The complexity of the enumeration is $\Theta\left(N|S|\right)$, so taking the model simulator into account, we have the global time complexity of $\Theta\left(N^2|S|\right)$.

### B.3   DP Solver

---

**Algorithm 3** DP Solver (basic version)

---

**Require:** $N, M, m, \texttt{ModelSimulator}_i\left(s_{\text{i-1}}, s_{\text{i}}, s_{\text{i+1}}\right)$
1: $\texttt{sol}(\cdot, \cdot, \cdot, \cdot, \cdot) \leftarrow \texttt{null}$
2: $\texttt{runtime}(\cdot, \cdot, \cdot, \cdot, \cdot) \leftarrow +\infty$
3: **for** $s_1 \in S$ **do**
4:    $\texttt{sol}(0, 0, 0, \texttt{null}, s_1) \leftarrow (s_1)$
5:    $\texttt{runtime}(0, 0, 0, \texttt{null}, s_1) \leftarrow 0$
6: **end for**
7: **for** $i = 1$ **to** $N$ **do**
8:    **for** $\text{mem}_{\text{curr}}, \text{mem}_{\text{peak}}, s_{i-1}, s_i$ **s.t.** $\texttt{sol}(i - 1, \text{mem}_{\text{curr}}, \text{mem}_{\text{peak}}, s_{i-1}, s_i) \neq \texttt{null}$ **do**
9:      **for** $s_{i+1} \in S$ **do**
10:       **if** $i = N$ **then**
11:         $s_{i+1} \leftarrow \texttt{null}$
12:       **end if**
13:       $\Delta\text{mem}_i, \text{residentMem}_i, \text{peakMem}_i, \text{runtime}_i \leftarrow \texttt{ModelSimulator}_i(s_{i-1}, s_i, s_{i+1})$
14:       $\text{mem}'_{\text{curr}} \leftarrow \text{mem}_{\text{curr}} + \text{residentMem}_i + \Delta\text{mem}_i$
15:       $\text{mem}'_{\text{peak}} \leftarrow \max\{\text{mem}_{\text{peak}}, \text{mem}_{\text{curr}} + \text{peakMem}_i\} + \text{residentMem}_i$
16:       $\text{runtime}' \leftarrow \texttt{runtime}(i - 1, \text{mem}_{\text{curr}}, \text{mem}_{\text{peak}}, s_{i-1}, s_i) + \text{runtime}_i$
17:       **if** $\text{mem}'_{\text{peak}} < m$ **and** $\text{runtime}' < \texttt{runtime}(i, \text{mem}'_{\text{curr}}, \text{mem}'_{\text{peak}}, s_i, s_{i+1})$ **then**
18:         $\texttt{sol}(i, \text{mem}'_{\text{curr}}, \text{mem}'_{\text{peak}}, s_i, s_{i+1}) \leftarrow \texttt{sol}(i - 1, \text{mem}_{\text{curr}}, \text{mem}_{\text{peak}}, s_{i-1}, s_i) + (s_{i+1})$
19:         $\texttt{runtime}(i, \text{mem}'_{\text{curr}}, \text{mem}'_{\text{peak}}, s_i, s_{i+1}) \leftarrow \text{runtime}'$
20:       **end if**
21:      **end for**
22:    **end for**
23: **end for**
24: $s_{\text{optimal}} \leftarrow \texttt{null}$
25: $\text{runtime}_{\text{optimal}} \leftarrow +\infty$
26: **for** $\text{mem}_{\text{curr}}, \text{mem}_{\text{peak}}, s_N$ **s.t.** $\texttt{sol}(N, \text{mem}_{\text{curr}}, \text{mem}_{\text{peak}}, s_N, \texttt{null}) \neq \texttt{null}$ **do**
27:    **if** $\texttt{runtime}(N, \text{mem}_{\text{curr}}, \text{mem}_{\text{peak}}, s_N, \texttt{null}) < \text{runtime}_{\text{optimal}}$ **then**
28:      $s_{\text{optimal}} \leftarrow \texttt{sol}(N, \text{mem}_{\text{curr}}, \text{mem}_{\text{peak}}, s_N, \texttt{null})$
29:      $\text{runtime}_{\text{optimal}} \leftarrow \texttt{runtime}(N, \text{mem}_{\text{curr}}, \text{mem}_{\text{peak}}, s_N, \texttt{null})$
30:    **end if**
31: **end for**
**Ensure:** $s_{\text{optimal}}$

---

The basic version of the DP solver is introduced in the main paper, while the pseudo-code is shown in Algorithm 3. Here we use $\texttt{ModelSimulator}_i\left(s_{\text{i-1}}, s_{\text{i}}, s_{\text{i+1}}\right)$ to represent the simulation of the $i$-th segment, while the result $\Delta\text{mem}_i, \text{residentMem}_i, \text{peakMem}_i, \text{runtime}_i$ denotes the delta memory after forward segment $i$, the resident memory of layer $i$, the peak memory during the forward and backward of segment $i$, and the total runtime of segment $i$, respectively.

For this basic version of DP solver, the state complexity is $\mathrm{O}\left(m^2 N|S|^2\right)$, while the transfer complexity is $\Theta\left(|S|\right)$. On this basis, we attempt to do some pruning to improve its efficiency.

We first notice that the optimization scheme $s_i, s_{i+1}, s_{i+2}$ are only used in the running segment simulation during transfer. However, the simulation for segment $i + 1$ only requires $s_{i+1}$ and part of $s_i, s_{i+2}$. Specifically, for the previous layer ($i$) and the next layer ($i + 2$), we only care about three communication-related switches: hidden offloading, lazy-prefetching, and parameter partition. Therefore, we change $s_i, s_{i+1} \in S$ to $r_i, r_{i+1} \in R$ in the DP state, where $R$ denotes the set of combinations of these three switches. Similarly, we only enumerate $r_{i+2} \in R$ for transfer because it is enough for the simulation of segment $i + 1$. In addition, we need to extend $r_{i+1}$ to $s_{i+1}$ by another enumeration when transferring. In this way, the state complexity can be optimized to $\Theta\left(m^2 N |R|^2\right)$, while the transfer complexity remains $\Theta\left(|S|\right)$ because the time complexity of enumerating $r_{i+2}$ and enumerating $s_{i+1}$ extended by $r_{i+1}$ are $\Theta\left(|R|\right)$ and $\Theta\left(\frac{|S|}{|R|}\right)$, respectively.

Besides, we can prune the non-optimal sub-solutions to improve efficiency further. In particular, for any two sub-solutions with the same $i, r_i, r_{i+1}$, if one has a lower $\text{mem}_{\text{curr}}$, a lower $\text{mem}_{\text{peak}}$, and a shorter runtime, then the other one will be pruned because it is definitely not an optimal substructure of the global optimal solution. In practice, this pruning can be easily implemented by a 2-dimensional scanning with the time complexity of $\Theta\left(m^2\right)$ for any given $i, r_i, r_{i+1}$.

Under the first pruning, the time complexity of the DP solver is reduced to $O\left(m^2 N |S||R|^2\right)$, where $|S| = 64, |R| = 8$. For the experiments in this paper, we always take $m = 128$. Although the second pruning does not improve the time complexity, it can reduce more than 99% of the DP states in actual runs, which greatly improves the solver's efficiency.

Based on the basic DP algorithm and the two prunings described above, we have the pseudo-code of our pruned DP solver shown in Algorithm 4.

## C  Experimental Settings

### C.1  Model Size

The model sizes used in the experiment are shown in Table 2. The parameters ($N$, $d_{\text{hidden}}$, $d_{\text{FFN}}$, and $N_{\text{head}}$) of BERT-1.8B, BERT-6B, and BERT-13B are referenced from T5-3B and T5-11B [35]. The parameters of BERT-100B are referenced from GPT3-175B [4].

Table 2: Different sizes of BERT in this paper.

| Size | #Parameter | $N$ | $d_{\text{hidden}}$ | $d_{\text{FFN}}$ | $N_{\text{head}}$ |
|------|-----------|-----|---------------------|------------------|-------------------|
| 1.8B | $1.84 \times 10^9$ | 48 | 1024 | 16384 | 32 |
| 6B | $6.68 \times 10^9$ | 48 | 1024 | 65536 | 128 |
| 13B | $1.38 \times 10^{10}$ | 48 | 2048 | 65536 | 128 |
| 100B | $1.02 \times 10^{11}$ | 56 | 12288 | 49152 | 96 |

### C.2  Hardware Configurations of Environment

$8 \times$ **2080Ti**: The GPU is NVIDIA GeForce RTX 2080 Ti (10 GB). The CPU is Intel(R) Xeon(R) CPU E5-2680 v4 @ 2.40GHz. The GPU number is 8, and the CPU core number is 56. CPU RAM size is 126GB. The PCI-E version is 3, and the bandwidth is 15.8GB/s. NVLink is not available.

$8 \times$ **A100**: The GPU is NVIDIA A100-SXM4-40GB. The CPU is Intel(R) Xeon(R) Platinum 8358 CPU @ 2.60GHz. The GPU number is 8, and the CPU core number is 128. CPU RAM size is 503GB. The PCI-E version is 4, and the bandwidth is 31.5GB/s. There are 12 fully connected NVLink 3.0.

$64 \times$ **A100**: 8 nodes of "$8 \times$ A100".

### C.3  Training Settings

As we mentioned in the main text, we implement CPU AdamW [24] based on Intel AVX instructions and use it to train our BERT models. More details of the implementation are stated in Section A.3.

In this paper, our profiler collects the runtime information in the first $S$ steps ($S \geq 10$). Then we run our solver to determine the optimization sequence of the model. After that, we skip $S$ steps to wait

**Algorithm 4** DP Solver (pruned version)

**Require:** $N, M, m, \texttt{ModelSimulator}_i\, (r_{\text{i-1}}, s_{\text{i}}, r_{\text{i+1}})$
1: $\texttt{sol}(\cdot, \cdot, \cdot, \cdot, \cdot) \leftarrow \texttt{null}$
2: $\texttt{runtime}(\cdot, \cdot, \cdot, \cdot, \cdot) \leftarrow +\infty$
3: $\texttt{optimState} \leftarrow \{\}$
4: **for** $r_1 \in R$ **do**
5:    $\texttt{sol}(0, 0, 0, \texttt{null}, r_1) \leftarrow ()$
6:    $\texttt{runtime}(0, 0, 0, \texttt{null}, r_1) \leftarrow 0$
7:    $\texttt{optimState} \leftarrow \texttt{optimState} \cup \{(0, 0, \texttt{null}, r_1)\}$
8: **end for**
9: **for** $i = 1$ **to** $N$ **do**
10:    **for** $(\text{mem}_{\text{curr}}, \text{mem}_{\text{peak}}, r_{i-1}, r_i) \in \texttt{optimState}$ **do**
11:       **for** $s_i \in \texttt{extended}(r_i)$ **do**
12:          **for** $r_{i+1} \in R$ **do**
13:             **if** $i = N$ **then**
14:                $r_{i+1} \leftarrow \texttt{null}$
15:             **end if**
16:             $\Delta\text{mem}_i, \text{residentMem}_i, \text{peakMem}_i, \text{runtime}_i \leftarrow \texttt{ModelSimulator}_i(r_{i-1}, s_i, r_{i+1})$
17:             $\text{mem}'_{\text{curr}} \leftarrow \text{mem}_{\text{curr}} + \text{residentMem}_i + \Delta\text{mem}_i$
18:             $\text{mem}'_{\text{peak}} \leftarrow \max\{\text{mem}_{\text{peak}}, \text{mem}_{\text{curr}} + \text{peakMem}_i\} + \text{residentMem}_i$
19:             $\text{runtime}' \leftarrow \texttt{runtime}(i - 1, \text{mem}_{\text{curr}}, \text{mem}_{\text{peak}}, r_{i-1}, r_i) + \text{runtime}_i$
20:             **if** $\text{mem}'_{\text{peak}} < m$ **and** $\text{runtime}' < \texttt{runtime}(i, \text{mem}'_{\text{curr}}, \text{mem}'_{\text{peak}}, r_i, r_{i+1})$ **then**
21:                $\texttt{sol}(i, \text{mem}'_{\text{curr}}, \text{mem}'_{\text{peak}}, r_i, r_{i+1}) \leftarrow \texttt{sol}(i - 1, \text{mem}_{\text{curr}}, \text{mem}_{\text{peak}}, r_{i-1}, r_i) + (s_i)$
22:                $\texttt{runtime}(i, \text{mem}'_{\text{curr}}, \text{mem}'_{\text{peak}}, r_i, r_{i+1}) \leftarrow \text{runtime}'$
23:             **end if**
24:          **end for**
25:       **end for**
26:    **end for**
27:    $\texttt{optimState} \leftarrow \{\}$
28:    **for** $r_i \in R$ **do**
29:       **for** $r_{i+1} \in R$ **do**
30:          **for** $\text{mem}_{\text{curr}} = 0$ **to** $m$ **do**
31:             **for** $\text{mem}_{\text{peak}} = \text{mem}_{\text{curr}}$ **to** $m$ **do**
32:                $\text{runtime}'_A \leftarrow \texttt{runtime}(i, \text{mem}_{\text{curr}} - 1, \text{mem}_{\text{peak}}, r_i, r_{i+1})$
33:                $\text{runtime}'_B \leftarrow \texttt{runtime}(i, \text{mem}_{\text{curr}}, \text{mem}_{\text{peak}} - 1, r_i, r_{i+1})$
34:                **if** $\texttt{runtime}(i, \text{mem}_{\text{curr}}, \text{mem}_{\text{peak}}, r_i, r_{i+1}) < \min\{\text{runtime}'_A, \text{runtime}'_B\}$ **then**
35:                   $\texttt{optimState} \leftarrow \texttt{optimState} \cup \{(\text{mem}_{\text{curr}}, \text{mem}_{\text{peak}}, r_i, r_{i+1})\}$
36:                **else**
37:                   $\texttt{runtime}(i, \text{mem}_{\text{curr}}, \text{mem}_{\text{peak}}, r_i, r_{i+1}) \leftarrow \min\{\text{runtime}'_A, \text{runtime}'_B\}$
38:                **end if**
39:             **end for**
40:          **end for**
41:       **end for**
42:    **end for**
43: **end for**
44: $\text{s}_{\text{optimal}} \leftarrow \texttt{null}$
45: $\text{runtime}_{\text{optimal}} \leftarrow +\infty$
46: **for** $\text{mem}_{\text{curr}}, \text{mem}_{\text{peak}}, r_N$ **s.t.** $\texttt{sol}(N, \text{mem}_{\text{curr}}, \text{mem}_{\text{peak}}, r_N, \texttt{null}) \neq \texttt{null}$ **do**
47:    **if** $\texttt{runtime}(N, \text{mem}_{\text{curr}}, \text{mem}_{\text{peak}}, r_N, \texttt{null}) < \text{runtime}_{\text{optimal}}$ **then**
48:       $\text{s}_{\text{optimal}} \leftarrow \texttt{sol}(N, \text{mem}_{\text{curr}}, \text{mem}_{\text{peak}}, r_N, \texttt{null})$
49:       $\text{runtime}_{\text{optimal}} \leftarrow \texttt{runtime}(N, \text{mem}_{\text{curr}}, \text{mem}_{\text{peak}}, r_N, \texttt{null})$
50:    **end if**
51: **end for**
**Ensure:** $\text{s}_{\text{optimal}}$

for the model warm-up. Finally, we measure the runtime for at least $S$ steps and reported the average result of the step time.

In the actual-run experiment, we set the memory constraint of the 2080Ti environment as $M = 8000$MB, and $M = 28$GB for A100. The memory constraints do not equal the GPU memory capacity because the actual memory occupation will be more than the memory usage due to memory fragmentation, memory cache, and other factors. We have a discussion about this problem in Section E.2 For all experiments in this paper, we set $m = 128$ as the default.

We use the half-precision float for all parameters and input tensors. Initial parameters and all input tensors are randomly generated. We fix the random seed and check the loss is invariant to ensure the correctness of our implementation. The sequence length is set to $512$, which is maximal for BERT.

We adopt BMTrain 0.1.8 and DeepSpeed 0.7.6 for our experiment. For a fair comparison, we turn on optimizer offloading and rematerialization for Megatron-DeepSpeed. In contrast, we do NOT introduce other optimizations such as tensor parallelism, pipeline parallelism, or FlashAttention for BOTH H3T and Megatron-DeepSpeed. We will leave the implementation of these optimizations and the comparison as our future work.

# D    Supplemental Experiments

## D.1    Memory Test

Table 3: The minimal memory cost of our framework and Megatron-DeepSpeed (unit: GB). We do not report the memory overhead of our automatic solvers because their memory overhead is always close to the upper bound due to their flexibility.

| Environment | | 8×2080Ti | | | | 8×A100 | | | | 64×A100 | | | |
|---|---|---|---|---|---|---|---|---|---|---|---|---|---|
| Model Size | | 1.8B | | 6B | | 6B | | 13B | | 13B | | 100B | |
| Global Batch Size | | 128 | 512 | 8 | 32 | 128 | 512 | 128 | 512 | 1024 | 4096 | 1024 | 2048 |
| BMTrain | Min Memory | **1.7** | **6.0** | **1.0** | **2.0** | **6.0** | **21.5** | **7.0** | **23.2** | **6.7** | **22.9** | **14.1** | **20.2** |
| Megatron -DeepSpeed | ZeRO-3 | 5.8 | - | - | - | 9.4 | - | 10.7 | - | - | - | - | - |
| | ZeRO-2 | 6.8 | - | - | - | 20.2 | - | 34.4 | - | 34.4 | - | - | - |

We test the minimal memory cost of the two frameworks under different settings, and the result is shown in Table 3. From the result, we can find that in most of our settings, our implementation supports the batch size at least $4\times$ larger than Megatron-DeepSpeed, which provides users more flexibility and convenience in training their models. We can also find that H3T can save $34.6\% \sim 70.7\%$ ($70.3\% \sim 80.5\%$) GPU memory compared with Megatron-DeepSpeed ZeRO-3 (ZeRO-2) with the same batch size and other settings. Furthermore, we are able to train larger models under limited hardware resources, such as training BERT-6B on $8\times$2080Ti.

Not only that, based on our implementation, we can train GPT-3-175B [4] on $64\times$A100 (sequence length is 512, and batch size is 512) while using only about 11.1 GB of memory per device at a minimum. This is really an exciting result because we would never have dreamed that a 175B-parameter model could be trained with such limited hardware, not to mention with such low GPU memory overhead. Unfortunately, we cannot train GPT-3-175B with a larger batch size or a sequence length due to the CPU RAM memory limitation of 512 GB. Currently, the step time of H3T training GPT-3 within this setting is about $40.8$ second. According to our observation, the bottleneck is now in communication but not computation, so if we can set the batch size and the sequence length larger, we will have a great increase in the training throughput. We will incorporate CPU RAM optimizations to solve this problem in our future work.

## D.2    End-to-end Experiment on SuperGlue

We conduct end-to-end training on four actual tasks from SuperGLUE [48] to check the soundness of our implementation. Specifically, we train BERT-large-cased [9] using BMTrain with H3T and PyTorch with APEX[5], respectively. When using PyTorch with APEX, we adopt mixed precision

---
[5]`https://github.com/NVIDIA/apex`

Table 4: The experimental results on SuperGLUE.

| Toolkits | BoolQ | CB (acc. / F1) | RTE | WiC |
|---|---|---|---|---|
| BMTrain w/ H3T | 75.12 | 94.64 / 96.17 | 71.07 | 72.50 |
| PyTorch w/ APEX | 67.34 | 94.64 / 95.91 | 71.48 | 72.73 |

training to ensure the fairness of comparison. The experimental result is shown in Table 4. From the result, we can find that our system can achieve comparable results with PyTorch on all four tasks, which greatly confirms the correctness of our implementation.

## D.3 Energy Test

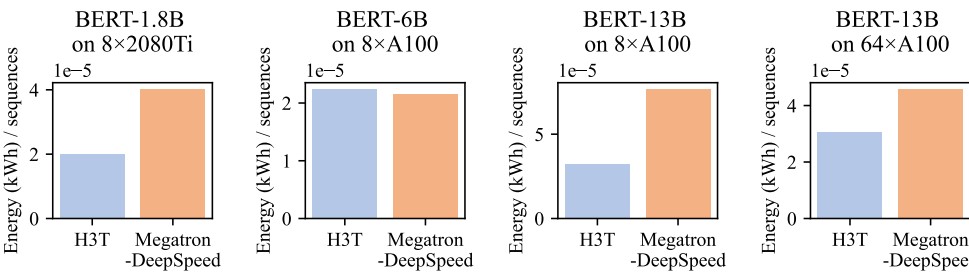

Figure 10: The energy cost (kWh) per sequence of H3T and Megatron-DeepSpeed. Here we use our DP solver for H3T and use the ZeRO-2 mode for Megatron-DeepSpeed.

We evaluate the energy cost of our framework and the baseline, Megatron-DeepSpeed, and the result is shown in Table 10. From the table, we find that our framework achieves lower energy cost with Megatron-DeepSpeed in most cases, except for training the 6B model on $8 \times$ A100, where we also achieve a comparable result. This result uncovers that our framework is more energy-saving than Megatron-DeepSpeed.

## D.4 Solver Efficiency Test

Table 5: The solver runtime within different settings. The environment is 64×A100. The runtime of the DP solver is influenced by many variables because of the pruning, while the other two are only influenced by the model size. Here $M$ denotes the memory constraint.

| Setting | | | Solver Runtime (min) | | |
|---|---|---|---|---|---|
| Model Size | Batch Size | $M$ (GB) | DP | Greedy | Rand. |
| 13B | 1024 | 25 | 12.0 | 0.6 | 0.1 |
| 100B | 1024 | 25 | 4.7 | 0.9 | 0.2 |
| 13B | 4096 | 25 | 2.8 | 0.6 | 0.1 |
| 13B | 1024 | 100 | 13.9 | 0.6 | 0.1 |

AI scientists and engineers sometimes change the model size and architecture in their works. For most people, this change does not always happen, so we do not need the solver to be as fast as possible, but the efficiency of the solver is still in demand. Therefore, we test the efficiency of the three solvers, and the result is shown in Table 5.

From the result, we can find that the random solver and greedy solver run significantly faster than the DP solver, although the minute-level runtime of the DP solver is quite acceptable compared to the training time of the large-scale model. The DP solver owes its acceptable efficiency to the two prunings, especially the second one, which reduces more than $99\%$ of the DP states. In contrast, DP without these two prunings will take over a day or even a month to get the solution, which is greatly unacceptable.

However, this is not the best for the DP solver. For deadline reasons, we use Python to implement the serial version of the three solvers. In the future, we will port them to C++ and optimize them using multithreading and parallelism techniques. In this way, we expect the DP solver to gain at least another $100\times$ improvement in efficiency and be much more senseless in the real application.

## D.5   Case Study

Table 6: A case study of H3T's optimization in different simulated cases.

Default settings: BERT-1.8B, batch size = 128, $8\times2080$Ti, $M = 8$, DP solver.

| Simulated case | Switch-on ratio of | | | |
|---|---|---|---|---|
| | parameter partition | lazy-prefetching | Rematerialiaztion | offloading hidden state |
| / | 90% | 8% | 92% | 21% |
| PCI-E speed $\times0.1$ | 0% | 0% | 98% | 0% |
| FPU speed $\times0.1$ | 88% | 90% | 88% | 81% |
| M$\leftarrow$25 | 50% | 50% | 65% | 56% |
| Solver$\leftarrow$Greedy | 0% | 0% | 100% | 0% |

We show a case study in Table 6 to intuitively show the ability of our automatic solvers. From the result, we find that if we manually slow down the PCI-E communication, the solver can adaptively switch on fewer communication-aware optimizations and more computation-aware optimizations. The opposite situation occurs if we manually slow down the computation. When the memory constraint becomes relaxed, the solver reduces the switch-on ratio of all four switches. These results show that the DP solver is well adapted to the actual environment to customize the optimization scheme. Besides, we find the greedy solver does not offer as much flexibility as the DP solver in provisioning multiple optimization switches, although it still performs well in most cases of the simulation and actual runs.

# E   Discussion

## E.1   Scalability of H3T

In this part, we discuss the scalability of H3T. Although H3T is based on the solvers and optimization switches introduced above and implemented on BMTrain for Transformer training in this paper, the framework and the idea of H3T can be applied in many other scenes. An illustration is shown in Figure 11.

**Optimization techniques and the solver**. Scholars and programmers can introduce any optimization or parallelism techniques and design their own solver for the specified optimization switches. The switches do not have to be binary and layer-level as they are in this paper, as long as they can be formalized as several variables. For example, we can have a global switch of pipeline parallelism, or we can also have some continuous optimization hyper-parameters, e.g., the ratio of tensor offloading. Then, the design of the solver highly depends on the optimization switches. To support different switches and hyper-parameters, we may need a different solver from that we proposed in this paper. Despite this, we argue that the three solvers designed in this paper work well for most discrete binary layer-level optimization switches.

**Models**. H3T is designed for Transformer-based models but can also be applied to other neural models, which can be formalized into the sequential model. Since the structure of sequential models and transformer-based models is becoming more and more widely used in recent years, we argue H3T has good generality and compatibility in the present and future.

**Toolkits**. In this paper, we implement H3T on BMTrain. Scholars and programmers can also implement H3T on other toolkits like DeepSpeed, etc.

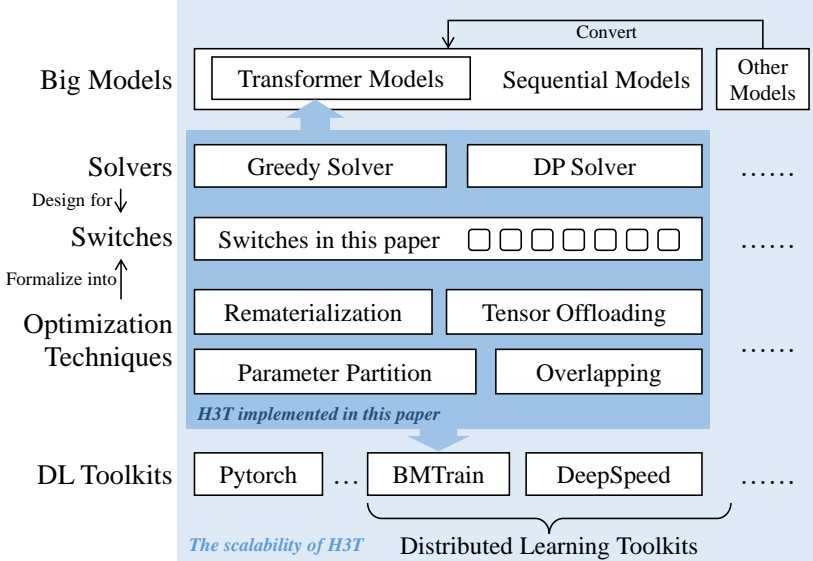

Figure 11: The scalability of H3T.

## E.2 Limitation of H3T

Although H3T performs well in our experiment and has good scalability, it still has some limitations. This section discusses the limitations of H3T from two perspectives: memory and model structure.

**Memory**. In this paper, we follow most existing works [2, 3, 18, 20] and defaultly use 'memory' to represent the memory allocation on GPU. Although this setting works well in most cases, it still brings two limitations.

(1) The memory allocation does not equal the actual memory usage. In actual runs, the GPU memory is usually managed by the underlying DL frameworks like PyTorch, TensorFlow, etc. And memory usage is usually greater than memory allocation due to many complex reasons, such as cache and memory fragmentations. Therefore, the user must set an $M$ lower than the actual GPU memory capacity to ensure H3T's proper running. Moreover, this also means H3T cannot perfectly estimate the memory usage, which leaves space for further improvement of our solver. One solution is to implement the memory pool management module in-house to gain absolute control over memory management [15]. To tackle this, we will develop a memory management module for absolute control over the GPU memory in the futures.

(2) We only focus on the GPU memory and neglect the CPU memory. This does not matter in most cases because the CPU RAM is usually large enough for all parameters, gradients, and optimizer states. However, when we train GPT-3-175B on $64 \times$ A100, the CPU RAM runs out of memory if we set the batch size as $1024$ or larger. This result reveals that CPU memory management is also in high demand for training extremely big models. Some researchers attempt to offload tensors from CPU RAM to the secondary storage (e.g., NVMe SSD) [37, 45], which is also our future direction to resolve this problem.

**Model structure**. As discussed in the scalability part, H3T supports any models that can be converted into sequential models. Nevertheless, there is one exception, namely, the models with dynamic structures, because we cannot search for a fixed optimization scheme for the dynamic model by pre-computing. This is a common problem with this kind of pre-computed optimization [3, 18, 20, 16, 31], and some other real-time optimization frameworks may be a better choice for these dynamic models [19, 15]. However, it is almost impossible to introduce approaches with heavy computations (e.g., DP and MILP) for real-time optimization, so most of these works employ heuristics to achieve adequate performance. In this paper, we mainly focus on Transformer-based models that are mostly fixed structures, so it is a better choice to design a pre-computed optimization framework like H3T.

