# OpenReview forum: "H3T: Efficient Integration of Memory Optimization and Parallelism for Large-scale Transformer Training"
_NeurIPS.cc/2023/Conference — NeurIPS 2023 poster_

### Official Review · Reviewer_QoRR · 2023-06-29

**Soundness:** 3 good
**Presentation:** 3 good
**Contribution:** 2 fair
**Rating:** 6
**Confidence:** 4

**Summary:**

Large language models have made significant advancements in terms of accuracy and representativeness across various applications. However, their large memory consumption and intensive computation size pose critical challenges for existing hardware platforms.

This paper proposes a memory-latency co-optimization training system that enables more efficient DNN training by reducing both memory consumption and latency. Given that the optimization is an NP-hard problem, this paper studies three optimization schemes using different solver searching algorithms. The evaluation demonstrates that the proposed solver-based optimizations significantly outperform other state-of-the-art frameworks, achieving up to 4.3x speedup and 80.5% less memory overhead.

**Strengths:**

- Training transformer is an important and hot research topic.
- This paper is well-organized and easy to follow. It is also well-motivated.
- The proposed techniques generally make sense.
- The evaluations are thorough and show a significant reduction in memory usage and latency.

**Weaknesses:**

- The paper lacks clarity on the contribution and requires further clarification on some technical details. The main contribution proposed in this paper is in Section 3.3.2, while there are limited details about these optimizations in the main paper, especially for the DP solver.
- To improve the understanding of the optimizations, it would be helpful to show a breakdown analysis of which specific optimization (e.g., re-computation, parallelism) contribute more to memory and latency reduction.
- Some figures have font sizes that are too small to read, such as Fig. 2.
- It would be clearer to explain the major difference between ZeRO-2 and ZeRO-3.

**Questions:**

- Further clarification of the memory reduction ratio would be helpful. Does the claim of 80.5% less memory consumption include the model size?
- What is the actual overhead (in terms of time) for the Brute-Force solution?
- Is the proposed solution compatible with layer reordering [1]?

[1] MCUNetV2: Memory-Efficient Patch-based Inference for Tiny Deep Learning

**Limitations:**

Yes

---

> ### Author Rebuttal · Authors · 2023-08-05
>
> Thanks for your thorough and contributive review. Here are the replies to your questions:
>
> 1. There are limited details about the proposed solvers, especially the DP solver.
>
> We are sorry for the lack of details in the main paper. But actually, we show many details about our solvers in the appendix (section B), especially for the DP solver (section B.3). In section B.3, we use nearly one page of text and nearly two pages of pseudo-code to introduce the details of the DP solver. We have to put them in the appendix due to the space limitation. This also follows some previous works like [1], which also describes the main idea in the main paper and puts the details in the appendix (this work is accepted by NeurIPS 2021).
>
> 2. To improve the understanding of the optimizations, it would be helpful to show a breakdown analysis of the optimization techniques.
>
> Thank you for your contributive suggestion! And we will seriously consider this suggestion in the revision. In this paper, we also put much effort into improving the understanding of the optimizations. For example, we show the timing diagram of each optimization switches in the appendix (section A). We also conduct a case study in section D.5 to show the characteristics of different optimizations.
>
> 3. Some figures have font sizes that are too small.
>
> We are sorry for the too-small font sizes, but we have to do so due to the space limitation. Once the paper accepts, we will have one more page, and we will make the font sizes larger in the revision.
>
> 4. It would be clearer to explain the major difference between ZeRO-2 and ZeRO-3.
>
> Thanks for the advice.
> ZeRO has three levels, and their main difference is: (1) ZeRO-1 distributes optimizer states; (2) ZeRO-2 distributes optimizer states and gradients; (3) ZeRO-3 distributes optimizer states, gradients, and parameters.
> We apologize that we misestimated how much people know about the ZeRO, and we will supplement its description in the revision.
>
> 5. Further clarification of the memory reduction ratio would be helpful. Does the claim of 80.5% less memory consumption include the model size?
>
> We are sorry for the ambiguity. The 80.5\% memory reduction occurs when training the 13B model with the batch size of 1024 on 64 × A100. In this case, Megatron-DeepSpeed allocates 34.4 GB of memory, and our framework only allocates 6.7 GB, so we have the memory reduction ratio equals (34.4-6.7)/34.4 = 80.5\%. This is the largest reduction ratio we observe in our experiments, and all the comparisons are across the same setting. The memory test result is shown in the appendix (section D.1, Table 2). Please refer to the appendix for more details.
>
> 6. What is the actual overhead (in terms of time) for the Brute-Force solution?
>
> This question is hard to answer because the actual overhead of Brute-Force is too high to estimate. The theoretical time complexity of Brute-Force is Θ(|S|^N × N), which means even for the 1.8B model (with N=48 layers), Brute-Force will take over 1.89 × 10^117 calculations to get the result. Even if 10^12 calculations can be done per second, we still need nearly 6 × 10^97 **years** to get the result. That is why we say this question is hard to answer, and this also demonstrates the value of our well-designed search algorithm.
>
> 7. Is the proposed solution compatible with layer reordering?
>
> The proposed DP solver does not support layer reordering because this solver is only designed for ZeRO, tensor offloading, rematerialization, and overlapping. However, as we discuss in the appendix (section E.1, line 168~176), Scholars and programmers can introduce any optimization or parallelism techniques and design their own solver for the specified optimization switches.
>
> Due to the space limitation, we have to move many important information and details to the appendix, and we are very sorry for your poor reading experience.
> However, we believe this paper is contributing, and our work can greatly promote the development of the AI community and LLM techniques. We would be grateful if you could reconsider your score!
>
> Thanks again for your careful review.
>
>
> [1] Olivier Beaumont, Lionel Eyraud-Dubois, and Alena Shilova. Efficient combination of rematerialization and offloading for training dnns. In Proceedings of NeurIPS, volume 34, pages 23844–23857, 2021.

---

> > ### Comment · Reviewer_QoRR · 2023-08-19
> >
> > Thank the author for clarifying my questions. It helps solve part of my concerns. I raised the final rating accordingly.

---

### Official Review · Reviewer_xJUr · 2023-07-03

**Soundness:** 2 fair
**Presentation:** 3 good
**Contribution:** 3 good
**Rating:** 4
**Confidence:** 4

**Summary:**

For efficient training of large language models, how parallelism and memory optimization are jointly configured is essential. Since the search space is too broad to explore, each has been studied separately. However, the separate search for memory optimization and parallelism could lead to a suboptimal solution. Therefore, efficiently exploring the joint search space and finding the optimal solution is a significant research problem.

This paper addresses it by proposing a novel system: H3T (High-Throughput Transformer Training). H3T incorporates memory optimizations (i.e., rematerialization and offloading) and parallelism strategies (data, model, and pipeline) into a single abstraction, which the authors defined as an optimization switch. Then, H3T automatically finds the best solution under the constraints (e.g., memory capacity) with a dynamic programming solver. The evaluation results show that the H3T solver can find better solutions than manual search results. Even more, their training performances (step time) are faster than Megatron-DeepSpeed.

**Strengths:**

As the authors pointed out, H3T is the first attempt to automatically integrate memory optimization and parallelism strategy for large Transformer-based models.

Moreover, the authors conducted various experiments, including memory tests, e2e evaluation, energy tests, and solver efficiency tests. The solve efficiency results show that H3T can find solutions in less than 20 minutes even for the DP solver.

Two typos found:
- Alph => Alpa in line 137
- $P_i$ and $ \nabla P_{i} $ is => $P_i$ and $ \nabla P_{i} $ are in line 191

**Weaknesses:**

Although the authors say that H3T covers model parallelism including pipeline parallelism, the paper does not explain how pipeline parallelism is incorporated. It looks like H3T is an automatic optimizer for tensor (parameter) parallelism and memory optimizations.

Furthermore, I think the experiments could be enhanced.
- Since H3T is an automatic optimizer, it is more plausible when adding a comparison with other automatic optimizers. For example, compared to Alpa or Unity, can H3T operate better?
- Did the Megatron-DeepSpeed results use the pipeline and tensor parallelism that Megatron provides? I expect that Megatron can perform much better for the 13B model without using the ZeRO optimizer (or only ZeRO-1 optimizer).


**Questions:**

Q) The paper highlights that **H3T can train a GPT-3 175B on 64xA100 40GB with a batch size of 512 and using only about 11 GB of memory per GPU**. Could you also give the step time of training the GPT-3 175B model? According to a Megatron paper [1], we can train the GPT-3 175B model on 64xA100 80GB with a notable step time of 13.75 seconds with a batch size of 64. Although it uses a different A100 cluster, each of which has 80GB memory capacity, comparing the values would be beneficial because the prices of A100 40GB and A100 80GB do not differ much (e.g., AWS p4d.xlarge is 32.77$ per hour and p4de.xlarge is 40.96 per hour). It could be much cheaper when using A100 80GB if possible.

Q) Does Megatron-DeepSpeed includes the latest Megatron [1] that supports selective recomputation and sequence parallelism?

Q) For the BERT-13B result with 64xA100 GPUs, why ZeRO-3 fails though ZeRO-2 succeeds? To my knowledge, ZeRO-3 is a more aggressive memory-saving approach than ZeRO-2.

Q) Do BMTrain and Megatron-DeepSpeed use FlashAttention?

Q) Is there any reason why $d_{FFN}$ is much bigger than $d_{hidden}$? Only the 100B model follows the common scaling ($d_{FFN} = 4  d_{hidden}$). For the others, it seems unnatural that $d_{FFN}$ is larger than $16  d_{hidden}$.

Reference

[1] Korthikanti, Vijay Anand, et al. "Reducing activation recomputation in large transformer models." Proceedings of Machine Learning and Systems 5 (2023).

**Limitations:**

 No potential negative societal impact exists in the paper. The authors adequately addressed the system's limitations in Appendix.

---

> ### Author Rebuttal · Authors · 2023-08-05
>
> Thanks for your thorough and contributive review. Here are the replies to your questions:
>
> Q1: Although the authors say that H3T covers model parallelism including pipeline parallelism, the paper does not explain how pipeline parallelism is incorporated.
>
> A1:
> We are sorry for your misunderstanding, but we only mention pipeline parallelism in the appendix (section E.1, line 171). In this paragraph, we mean anyone can introduce any optimization or parallelism techniques and design their own solver for the specified optimization switches, including pipeline parallelism. For the H3T implemented in this paper, we mainly focus on ZeRO, which does not normally coexist with pipeline parallelism. We leave the TP and PP version H3T as our future work.
>
> Q2: There may be a comparison with other automatic optimizers like Alpa or Unity.
>
> A2:
> Although H3T, Alpa, and Unity are all automatic optimizers, they are very different types of automatic optimizers. H3T mainly focuses on layer-level optimizations, such as tensor offloading, rematerialization, and ZeRO. In contrast, Alpa and Unity are kernel-level and operator-level optimizers, which means their granularity is finer, but they struggle to do a better job of optimizing for the structure of transformer-based models. Therefore, we argue that this comparison is unfair, and its priority is not high in our paper. We still sincerely thank you for your suggestion, and we will seriously consider it as a supplemental experiment in the revision.
>
> Q3: Did the Megatron-DeepSpeed results use the pipeline and tensor parallelism that Megatron provides?
>
> A3:
> No. As we mentioned in A1, we mainly focus on ZeRO in this paper, and we will leave the exploration of other parallelism methods in our future work. Specifically, we will design a TP and PP version of H3T and try to beat Megatron-DeepSpeed with TP and PP.
>
> Q4: Could you also give the step time of training the GPT-3 175B model?
>
> A4:
> The step time of training GPT-3 175B is about 40 seconds with a batch size of 512.
>
> Q5: Megatron can train the GPT-3 175B model on 64xA100 80GB with a notable step time of 13.75 seconds with a batch size of 64.
>
> A5: Memory capacity is a very important reason, but the config is more critical. As we mentioned in A2, we want our comparison to be as fair as possible on basic optimization techniques. Therefore, we employ ZeRO as the parallelism scheme for both H3T and Megatron-DeepSpeed.
> Besides, our throughput (512 / 40 = 12.8) is still significantly better than the result reported by Megatron's paper (64 / 13.75 = 4.7). This result may provide a comparison between H3T and a well-configured Megatron.
>
> Q6: Does Megatron-DeepSpeed includes the latest Megatron that supports selective recomputation and sequence parallelism?
>
> A6: No. We will test the selective recomputation and update the result in the revision. Thanks for this advice. Similar to TP and PP, for fairness, we do not test sequence parallelism in our experiment.
>
> Q7: For the BERT-13B result with 64xA100 GPUs, why ZeRO-3 fails though ZeRO-2 succeeds? To my knowledge, ZeRO-3 is a more aggressive memory-saving approach than ZeRO-2.
>
> A7: Because the ZeRO-3 version is stuck in this experiment. We are not sure whether there is a bug or if we do not run it correctly. But referring to the results on the 8 × A100, we believe that ZeRO-3 will perform very poorly even if it is able to run, so this does not affect the overall conclusions of our experiments.
>
> Q8: Do BMTrain and Megatron-DeepSpeed use FlashAttention?
>
> A8: No. Both of them do not use FlashAttention. So the comparison is fair.
>
> Q9: Is there any reason why $d_{FFN}$ is much bigger than $d_{hidden}$? Only the 100B model follows the common scaling ($d_{FFN} = 4 d_{hidden}$). For the others, it seems unnatural that $d_{FFN}$ is larger than $16 d_{hidden}$.
>
> A9: The parameters of these models are mainly referenced to T5 (3B and 10B). The diversity of settings could also better validate the generalizability of H3T capabilities.
>
> Besides, we thank you for pointing out the typos (and the detail problems, such as Q3, Q4, Q6, Q7, Q8, and Q9). We will fix the typos and supplement these details in the revision.
>
> We understand that you would like to see more comparative experiments with other toolkits or Megatron-DeepSpeed with other settings.
> We appreciate your constructive suggestions, but on the one hand, we prefer to present the most critical comparisons (i.e., fair comparisons using the same underlying optimization methods) in the limited paper space, as this is more indicative of the strengths of our underlying implementation and automated optimization solver. On the other hand, the config of Megatron-DeepSpeed is very complex, and we believe this is why we need an automatic solver to decide the appropriate optimization config. Besides, although H3T implemented in this paper only consider ZeRO as the parallelism solution, it still beats a well-configured Megatron-DeepSpeed (according to the throughput comparison in A5), which reflects BMTrain with H3T is already a better toolkit than Megatron-DeepSpeed. As mentioned in A1, we will further develop the H3T to support TP, PP, and other optimization and parallelism techniques and make it a better toolkit.
>
> In summary, we argue that our experiments are convincing enough to support the contribution of this paper. We would be grateful if you could reconsider your score! (We would like to take the liberty of confirming with you that you know the review is on a 10-point scale :).)
>
> Thanks again for your careful review.

---

> > ### Comment · Reviewer_xJUr · 2023-08-14
> >
> > Thank you for the thorough responses.
> > Most of my questions are well-clarified, but I have a single additional question.
> > Is the sequence length of the GPT-3 175B model (with 40 seconds of step time) also tested with 512?
> > If so, comparing "token" throughputs is reasonable than comparing "batch" throughputs and it seems unlikely to say **significantly better** even considering the hardware difference.
> > In other words: $$ \frac{\textrm{BatchSize} \times \textrm{SequenceLength}}{\textrm{StepTime}} $$
> >
> > I have slightly raised the score, but I am cautious about raising more. In my opinion, since the authors tackle the problem of memory optimization and parallelism, comparing the solution with state-of-the-art techniques is important. For example, both selective recomputation (including FlashAttention) and sequence parallelism notably improve memory usage and training throughputs.

---

### Official Review · Reviewer_khhX · 2023-07-05

**Soundness:** 3 good
**Presentation:** 3 good
**Contribution:** 3 good
**Rating:** 6
**Confidence:** 3

**Summary:**

This paper presents a framework to automatically find an optimized integration of memory and parallelism for training large language models (LLMs). To this end, this paper first formulates the memory optimization strategies and parallelism techniques, and then solves the equation to find the best configuration for the training of LLMs. Using three different solvers, this paper shows a significant boost in terms of training speed and memory compared to the DeepSpeed framework across different models.

**Strengths:**

-- The main strength of this paper is its results on training LLMs. It shows a significant improvement compared to DeepSpeed.
-- The main idea of finding the best memory optimization strategy and parallelism technique automatically is interesting and of significant importance since the models are becoming larger.

**Weaknesses:**

-- The main concern of mine is related to the configuration used for training using DeepSpeed. DeepSpeed supports different memory optimization strategies and parallelism techniques which can be set manually via its configuration file. However, Table 1 of this paper shows that DeepSpeed fails to fit some models. I don't understand what could be the reason. My understanding of DeepSpeed is that with a proper config, it can fit the model at the cost of speed. Moreover, increasing the batch size shouldn't increase the memory as long as it reflects as gradient accumulation steps as discussed in PipeDream [25]. Therefore, my suggestion to the authors is to include the details of memory strategy and parallelism technique used in Table 1. Without such details, it is hard to comment on the significance of this paper.

**Questions:**

1- My main question is about the config file of DeepSpeed for the results in Table 1 (see https://huggingface.co/docs/accelerate/usage_guides/deepspeed). Without the details of DeepSpeed configuration, it is hard for me to evaluate the results of this paper. As a result, I will go with a midpoint score for now. Once I see the config file, I'll decide to increase or decrease my score.
2- Have you also compared your work with FSDP? How do you perform compared to this framework?

---

> ### Author Rebuttal · Authors · 2023-08-05
>
> Thank you for your appreciation of our work. We notice that your primary concern is about the configuration of DeepSpeed used in our experiment. We briefly describe it in the appendix (section C.3, line 108): "For fairness, we turn on optimizer offloading and rematerialization for Megatron-DeepSpeed." We show the key part of the DS config as follows, and we will release the detailed config of DeepSpeed in the revision.
>
> ```json
> "zero_optimization": {
>   "stage": 2,
>   "offload_optimizer": {
>     "device": "cpu",
>     "buffer_count": 4,
>     "pipeline_read": false,
>     "pipeline_write": false,
>     "pin_memory": true
>   }
> },
>
> "fp16": {
>   "enabled": true,
>   "initial_scale_power": 12
> },
> ```
>
> It is worth mentioning that according to our test, the parameter offloading is automatically switched on if we switch on optimizer offloading. Therefore, the comparison between our framework and Megatron-DeepSpeed is generally fair.
>
> Another key point we would like to explain is that, we do not switch on tensor parallelism (TP) and pipeline parallelism (PP) in our experiments. The main reason is that our implementation of H3T in this paper does not involve TP and PP, so we think it is fair to switch off TP and PP for Megatron-DeepSpeed. In the future, we will design the solver for TP and PP schemes. We will also implement it and compare it with Megatron-Deepspeed's TP and PP versions.
>
> It is also a good idea to compare our framework with FSDP. We do not try it, but we believe it will be a great baseline as a supplement and make our experiments more solid. We will seriously consider this suggestion in the revision.
>
> Thanks again for your careful and contribute review.

---

> > ### Comment · Reviewer_khhX · 2023-08-18
> > **Rebuttal Comments**
> >
> > Thank you for your responses. Overall, the paper looks good and outperforms Megatron in terms of memory and latency. In fact, the results reported for H3T in this paper are better than what I get for CodeGen-7B model using DeepSpeed on GCP and 8 A100 40GB. I can also understand the concerns of other reviewers about experiments. As such, I keep my previous rating for this paper.

---

### Official Review · Reviewer_qHi5 · 2023-07-06

**Soundness:** 2 fair
**Presentation:** 2 fair
**Contribution:** 2 fair
**Rating:** 5
**Confidence:** 3

**Summary:**

This paper considers large model training and proposes a search algorithm for various memory optimizations.

**Strengths:**

Please see the "Questions" section.

**Weaknesses:**

Please see the "Questions" section.

**Questions:**

- Lines 89-91: In what sense are you using the terms "dynamic" and "static"? I think, according to the most common way these terms are used, the methods referenced in the paragraph starting in line 89 are all static.

- Line 105: "These two optimization strategies are independent". I think this statement may need to be changed or removed. Rematerialization and offloading are not necessarily independent in the sense that one can offload a tensor for predecessor node (in a computation graph) and then later use that tensor to rematerialize another tensor (e.g. output of a successor node). There is actually recent work that tries to optimizes the integrated search space for these two memory optimization techniques. Please correct me if I misunderstand what is meant by "independent".

- I'm confused about one thing related to the proposed algorithm. Does the algorithm optimize for parallelism? That is, does your search include different parallelism schemes or are you assuming a fixed parallelism scheme and search for other things given this fixed parallelism?

- The Megatron-DeepSpeed baseline in Table 1: Do you have a sense of why ZeRO-2 beats ZeRo-3?

- For ZeRO, are you running mixed-precision training? Perhaps more importantly, what precision is used for your method vs ZeRO?

Minor:
- no space between analysis and [26] in line 30
- Line 74, MM is short for?
- Is it convention to capitalize "Transformers"?
- Is this statement in line 85 correct: "Rematerialization is first proposed by Griewank et al. [12]"? I believe there are older references for rematerialization.

**Limitations:**

Please see the "Questions" section.

---

> ### Author Rebuttal · Authors · 2023-08-03
>
> Thanks for your thorough and contributive review. Here are the replies to your questions:
>
> 1. The misuse of "dynamic" in lines 89-91.
>
> Thanks for pointing out this mistake. Our point is that these approaches are based on static rules which cannot dynamically adapt to the changes in model structure and hardware resources. However, "dynamic" and "static" do have other meanings according to the most common way they are used. We will fix this problem in the revision.
>
> 2. Rematerialization and offloading are not independent.
>
> Here "independent" means they do not influence each other in general. In other words, their core ideas and logic do not conflict, so switching on either of them does not means we cannot switch on the other one. However, we also noticed this representation is ambiguous at your reminder. Thanks for pointing this out, and we will fix this in revision.
>
> 3. Does the algorithm optimize for parallelism?
>
> Our DP and greedy solver can optimize for parallelism, but in this paper, our implementation focuses more on the memory optimization solution. We have a discussion about this problem in the appendix (section E.1, lines 168~176). Please refer to the appendix for it.
>
> 4. Do you have a sense of why ZeRO-2 beats ZeRO-3?
>
> ZeRO-3 optimize more but brings more communications, which means ZeRO-3 may be slower than ZeRO-2 (at least not faster than ZeRO-2). Moreover, we guess the ZeRO-3 implementation of Megatron-DeepSpeed is sub-optimal, which also makes ZeRO-3 much slower.
>
> 5. What precision is used for your method vs ZeRO?
>
> We use float16 for both BMTrain with H3T and Megatron-DeepSpeed in our experiment. We think it is a fair competition between our proposed framework and the baseline.
>
> 6. Line 74, MM is short for?
>
> MM is short for Multi-Modal. Thanks for pointing this problem out, and we will supplement the full name in the revision.
>
> 7. There are older references for rematerialization than Griewank et al. [12].
>
> We apologize for our laxity, and thanks for pointing this out. Jose Grimm et al. do have another work about rematerialization in 1996. We will fix this in the revision.
>
> We also thank you for pointing out the other typos, and we will fix all of them in the revision.
>
> We argue this paper is contributive because it not only proposes a novel and scalable framework to solve the difficulty of training LLMs, but also implements an out-of-the-box version based on the open-source toolkit BMTrain, which is helpful for developers and researchers. We believe both of our contributions are valuable for the AI community and can greatly promote the development of AI and LLM techniques. We would be grateful if you could reconsider your score.
>
> We sincerely thank you again for your careful review.

---

> > ### Comment · Reviewer_qHi5 · 2023-08-17
> >
> > Thanks for the responses. Overall, I hold a favorable view of the contributions of this paper. At the same time, I agree with the point that Reviewer xJUr makes about the experiments.

---

### Decision · Program_Chairs · 2023-09-21

**Decision:**

Accept (poster)

**Comment:**

Paper proposes a system, High-Throughput Transformer Training (H3T), that incorporates memory optimizations and parallelism strategies into a single optimization problem that is then yields a solution under the constraints (e.g., memory) with a DP solver. Resulting solution is compared against an off the shelf, (near) state of the art solution and found to be outperforming it (Megatron-DeepSpeed).

Considering the proposal being the first attempt to automatically integrate memory optimization and parallelism strategy for LLM training and convincing results, the paper is ready to be presented to the wider research audience. That said, some of the reviewers raised concerns mostly revolving around the "generalization" of the proposal and lack of experiments that could give convincing evidence on this matter.

With the current form and content, the papers targeted impact area is limited to the set of experimental configs and tools included in the design, which is quite significant already. For wider adoption and recognition the authors could consider building on top of their current work, but the submission is, to my assessment, over the bar to start discussions along this line of research -- which is quite new and important.